# An efficient and scalable pipeline for epitope tagging in mammalian stem cells using Cas9 ribonucleoprotein

Pooran Singh Dewari[1,2], Benjamin Southgate[1,2], Katrina Mccarten[1,2], German Monogarov[3], Eoghan O'Duibhir[1,2], Niall Quinn[4], Ashley Tyrer[1,2], Marie-Christin Leitner[1,2], Colin Plumb[1,2], Maria Kalantzaki[1,2], Carla Blin[1,2], Rebecca Finch[1,2], Raul Bardini Bressan[1,2], Gillian Morrison[1,2], Ashley M Jacobi[5], Mark A Behlke[5], Alex von Kriegsheim[4], Simon Tomlinson[1,2], Jeroen Krijgsveld[3], Steven M Pollard[1,2]*

[1]Edinburgh Cancer Research United Kingdom Centre, University of Edinburgh, Edinburgh, United Kingdom; [2]MRC Centre for Regenerative Medicine , University of Edinburgh, Edinburgh, United Kingdom; [3]German Cancer Research Center, University of Heidelberg, Heidelberg, Germany; [4]Institute of Genetics and Molecular Medicine, Edinburgh Cancer Research United Kingdom Centre, University of Edinburgh, Edinburgh, United Kingdom; [5]Integrated DNA Technologies, Inc., Coralville, United States

**Abstract** CRISPR/Cas9 can be used for precise genetic knock-in of epitope tags into endogenous genes, simplifying experimental analysis of protein function. However, Cas9-assisted epitope tagging in primary mammalian cell cultures is often inefficient and reliant on plasmid-based selection strategies. Here, we demonstrate improved knock-in efficiencies of diverse tags (V5, 3XFLAG, Myc, HA) using co-delivery of Cas9 protein pre-complexed with two-part synthetic modified RNAs (annealed crRNA:tracrRNA) and single-stranded oligodeoxynucleotide (ssODN) repair templates. Knock-in efficiencies of ~5–30%, were achieved without selection in embryonic stem (ES) cells, neural stem (NS) cells, and brain-tumor-derived stem cells. Biallelic-tagged clonal lines were readily derived and used to define Olig2 chromatin-bound interacting partners. Using our novel web-based design tool, we established a 96-well format pipeline that enabled V5-tagging of 60 different transcription factors. This efficient, selection-free and scalable epitope tagging pipeline enables systematic surveys of protein expression levels, subcellular localization, and interactors across diverse mammalian stem cells.
DOI: https://doi.org/10.7554/eLife.35069.001

*For correspondence:
steven.pollard@ed.ac.uk

## Introduction

Defining the protein levels, subcellular localisation and biochemical interactions for the >20,000 protein-coding genes in the mammalian genome remains a formidable challenge. Ideally, these would be explored across diverse primary cells – rather than genetically transformed and corrupted cell lines. Large-scale projects, such as the human protein atlas, have performed systematic characterisation of antibodies against native proteins (*Thul et al., 2017*). However, there are inherent difficulties in discovering, validating and distributing high-quality antibodies that cover all species and key applications (e.g. immunoblotting, immunoprecipitation, ChIP-Seq and immunocytochemistry).

A complementary strategy is to use epitope tagging: the fusion of small peptide-coding sequences to a protein of interest (e.g. 3XFLAG, HA, V5, and Myc) (*Jarvik and Telmer, 1998*). In contrast to plasmid-driven cDNA overexpression, which creates artificially high-protein levels, the knock-in of

**eLife digest** Genes are often referred to as the blueprints of life. Understanding the role of the genes in human cells is one of the major goals of biology. Recent advances in gene editing technologies, such as CRISPR/Cas9, mean scientists can now edit or delete precise sections within human genes, similar to how we edit words in a document on a computer. This has made it possible to insert small sequences that encode specific "tags" into genes. This in turn means that when a protein is built following the instructions in the gene, the protein includes the tag too, making it easy to monitor.

Tags on proteins can help scientists understand what those proteins do by answering various questions, such as: where is the protein found in the cell? How much of the protein is there in each cell? Does this change as the cell matures? What does the protein interact with? Yet, more research could be done if the tagging process was made easier, quicker and more efficient.

Dewari et al. have now come up with an improved gene editing approach that enabled them to rapidly tag hundreds of proteins all at the same time, with efficiencies that were much higher than expected based on previous approaches. The strategy uses common "off the shelf" reagents that can be designed with a new user-friendly, web-based tool called "Tag-IN". Dewari et al. focused on optimizing their method in freshly grown stem cells, originally collected from mice and humans. They then went on to show the scalability and efficiency of this approach by tagging 60 different proteins in brain stem cells from mice.

Now, rather than being limited to a handful of genes of interest, scientists can explore large families of genes in a variety of mouse and human cells in a much quicker and more comprehensive manner. Also, working with stem cells that can be freshly collected from individuals rather than cells that have been grown in the laboratory for a long time will be more useful for biological and disease studies. In the long-term, more knowledge of how protein-coding genes work in different human cells will benefit patients as new drugs or therapeutic targets are discovered.

DOI: https://doi.org/10.7554/eLife.35069.002

small epitope tags to endogenous genes provides physiologically relevant levels. A small set of pre-validated tag-specific antibodies is then used across diverse downstream experimental applications. This has been a key tool in yeast studies, resulting in global characterization of core protein complexes and their extensive interaction networks (*Gavin et al., 2006*; *Krogan et al., 2006*). However, to date this approach has not been widely adopted in mammalian cells, primarily due to the poor efficiency of homologous recombination (HR). Repurposed programmable nucleases now provide a potential solution.

Formation of a site-specific DNA double-strand break (DSBs), will massively enhance HR-mediated repair. This has become straightforward with the discovery and application of clustered regularly interspaced short palindromic repeats (CRISPR) and CRISPR-associated (Cas) proteins as designer site-specific nucleases. CRISPR/Cas9 was uncovered as a form of microbial adaptive immunity (*Bhaya et al., 2011*) that has been repurposed for genome editing in mammalian cells (*Cong et al., 2013*; *Doudna and Charpentier, 2014*).

Cas9 is an RNA-guided endonuclease that binds complementary DNA via formation of a 20 bp RNA:DNA heteroduplex. Stable binding of Cas9/gRNA complex at the target site leads to activation of nuclease domains and formation of a double-stranded DNA break (DSB) (*Jinek et al., 2014*). DSBs are predominantly repaired through the error-prone non-homologous end joining (NHEJ) pathway, resulting in insertion or deletion mutations (indels) (*Sander and Joung, 2014*). Alternatively, at lower frequencies HR-mediated repair will occur.

Successful knock-in of tags, therefore, requires co-delivery of three ingredients to the mammalian cells: the Cas9 protein, a guide RNA, and a donor repair template (i.e. single or double-stranded DNA encoding the tag or reporter with homology arms). These CRISPR components are typically delivered via transient plasmid transfection or viral vectors. Bespoke targeting vector plasmids are usually constructed for delivery of large cargoes or conditional alleles. Selection strategies, such as flow cytometry or use of antibiotic resistance cassettes, are then used to enrich for edited cells.

Despite the successes of current approaches (*Dalvai et al., 2015*; *Savic et al., 2015*; *Mikuni et al., 2016*; *Xiong et al., 2017*), it is clear that many bottlenecks restrict widespread applications; (i) Production of tailored plasmids for each component can be tedious and time-consuming; (ii) Plasmid-based selection strategies introduce additional regulatory elements that can disrupt normal regulatory processes; (iii) Plasmid DNA, either Cas9/gRNA expression vectors or targeting vectors, can randomly integrate in the genome causing insertional mutagenesis or increasing risks of off-target cleavage (*Kim et al., 2014*; *Liang et al., 2015*); (iv) current strategies are often not readily scalable to enable routine exploration of large numbers of genes; (v) recovery of biallelic clonal lines is inefficient. Thus, there remains an unmet need for improved knock-in strategies that can work efficiently in primary mammalian cells, such as pluripotent (ES/iPS), multipotent (tissue stem cells) and cancer stem cells.

Improved efficiencies of CRISPR editing in mammalian cells have been demonstrated using recombinant Cas9 protein (*Kim et al., 2014*; *Ramakrishna et al., 2014*; *Bressan et al., 2017*). Cas9 protein is complexed with in vitro-transcribed (IVT) RNA, to produce a ribonucleoprotein (RNP) complex that can be then delivered into cells. The RNPs are short-lived and cleared by cells within 24–48 hr, reducing the risk of both formation of mosaic clones or off-target cleavage (*Kim et al., 2014*; *Lin et al., 2014*; *Zuris et al., 2015*; *Cameron et al., 2017*). However, RNP-assisted methods have mainly been used for gene knock-out (*Kim et al., 2014*; *Liang et al., 2015*), incorporation of point mutations (*Ma et al., 2017*; *Rivera-Torres et al., 2017*), or knock-in of restriction enzyme sites (*Lin et al., 2014*; *Schumann et al., 2015*). We and others have recently made use of RNP with in vitro-transcribed (IVT) sgRNAs for CRIPSR epitope tagging (*Bressan et al., 2017*; *Liang et al., 2017*), yet the efficiencies in the absence of selection are highly variable, and this approach cannot be scaled.

To avoid IVT, chemically modified ~100 nt-long sgRNAs can be synthesised (*Hendel et al., 2015*); however, these are prohibitively expensive, limiting applications. Alternatively, a two-part, chemically synthesised, short target-specific crRNA plus a longer generic tracrRNA can be used. This is cheaper (only the crRNA needs to be resynthesised for new targets) and has better performance (*Aida et al., 2015*; *Anderson et al., 2015*). This 'dual-RNA' approach has been advanced further by 'base, backbone and end' modifications of cr/tracrRNAs (*Rahdar et al., 2015*; *Kelley et al., 2016*) and also by the use of shorter and more effective modified cr/tracrRNAs (*Jacobi et al., 2017*). Modified synthetic cr/tracrRNAs are resistant to nuclease digestion, limit cellular immune responses, have greater stability, and therefore provide enhanced targeting efficiency (*Jacobi et al., 2017*).

Here, we explored whether RNPs with synthetic modified two-part guide RNA can enhance efficiency of epitope tagging in mammalian stem cells. We find that co-delivery of a Cas9 RNP (dual-synthetic RNA) with ~200 bp single-stranded oligodeoxynucleotides (ssODNs) supports highly efficient epitope tagging. This is achieved across a variety of stem cell types without any requirement for plasmids, selection steps, flow-cytometry-based enrichment, or IVT reactions. To provide a proof-of-principle, we developed a novel web-based design tool and demonstrate effective tagging in 96-well plate format. We demonstrate one application, by identifying Olig2 protein partners using immunoprecipitation-mass spectrometry (IP-MS).

## Results

### Cas9 protein complexed with synthetic cr/tracr RNAs enables highly efficient epitope tagging in neural and glioma stem cells

sgRNAs produced by IVT reactions can vary in quality and quantity and are prone to degradation, either during production and/or following delivery into cells. We therefore explored a synthetic modified two-part guide RNA system (annealed 36-mer crRNA: 67-mer tracrRNA) (*Anderson et al., 2015*; *Jacobi et al., 2017*).

Guide RNAs were designed to cut proximal to the stop codon in the 3' UTR of *Olig2* or *Sox2* (*Figure 1A*). The efficacy of custom synthetic modified RNAs (csRNAs) was compared to IVT-generated sgRNAs. RNA was complexed with recombinant Cas9 protein and transfected into an adult mouse neural stem (NS) cell line (ANS4), using an optimised nucleofection program. RNP was delivered together with a ~ 200 bp single-stranded DNA donor encoding the V5 tag, flanked with ~70 nucleotide homology arms (*Figure 1B*). After 5 days, cells were analysed using immunocytochemistry

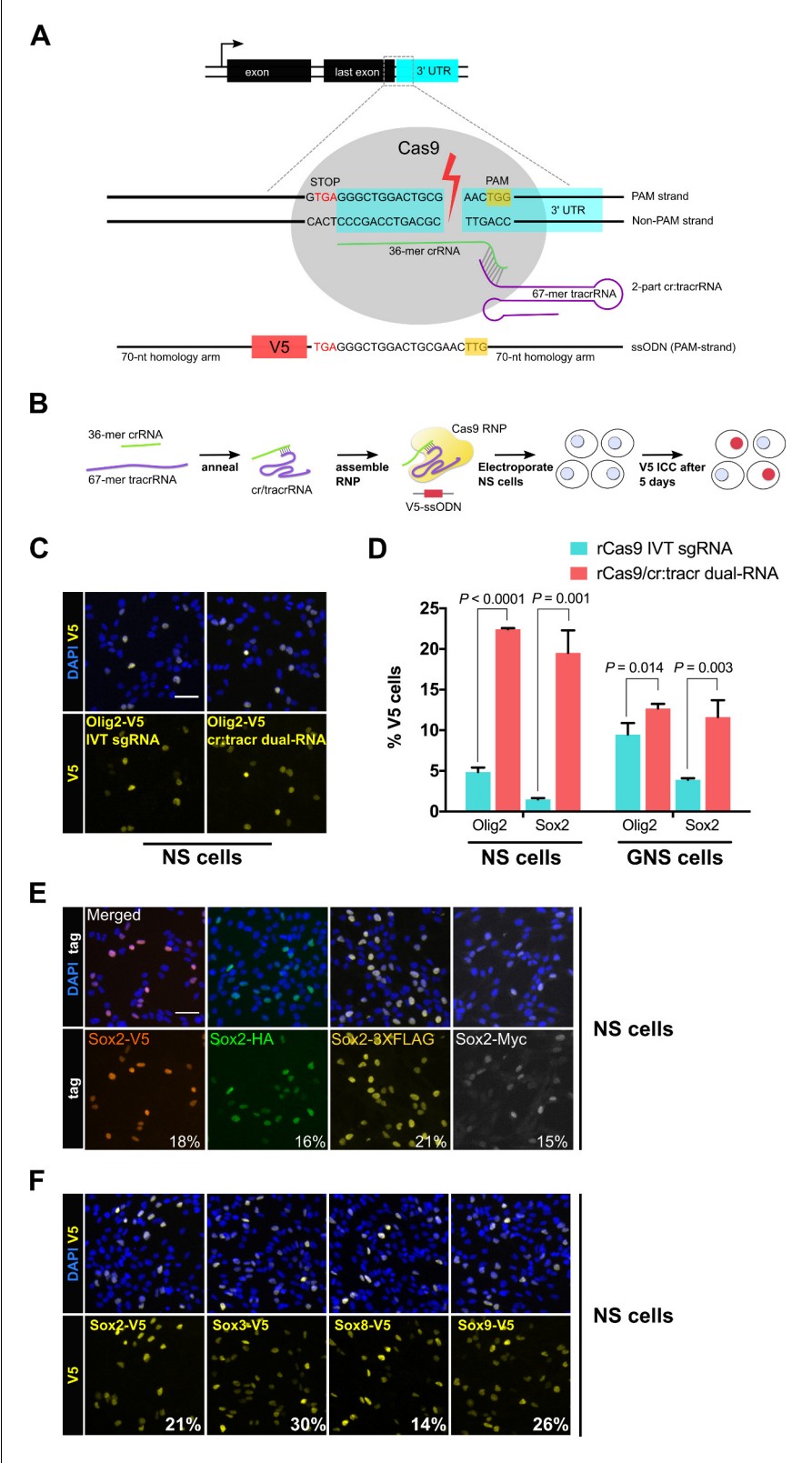

**Figure 1.** Cas9 protein in complex with synthetic cr/tracrRNAs enables highly efficient knock-in of biochemical tags in mouse neural and glioma stem cells. (**A**) Schematic representation of epitope knock-in strategy. A crRNA was designed against the 3'UTR of each target gene. Target site with double-stranded break is shown with Cas9 RNP (grey), PAM in yellow box, and single-stranded donor DNA that harbours PAM-blocking mutations and V5 tag coding sequence flanked by 70-mer homology arms on both sides. (**B**) Cas9 RNP complexes were assembled in vitro by incubation of recombinant

*Figure 1 continued on next page*

*Figure 1 continued*

Cas9 protein with either IVT sgRNA or synthetic two-part cr:tracrRNA and electroporated into NS cells. V5 ICC was used to quantify knock-in. (**C**) Representative ICC images for the detection of Olig2-V5 fusion protein in the bulk populations of transfected cells. (**D**) HDR-mediated insertion of V5 tag was determined by scoring V5-positive cells (%) in the bulk populations of transfected cells. Results from three independent experiments are shown for *Sox2* and *Olig2* V5 tagging using mouse neural stem (NS) and glioma-initiating neural stem (GNS) cells. Error bars indicate standard deviation values based on a minimum of two experiments, p-values were derived using unpaired t test. (**E**) ICC for *Sox2* gene epitope tagging at the C-terminus with V5, HA, 3XFLAG, or Myc epitope. Numbers represent percentage of tagged cells in the bulk population for each tagging experiment. (**F**) Representative bulk population V5 ICC images for Sox2, Sox3, Sox8, and Sox9 V5 knock-in are shown. Average knock-in efficiency from two independent experiments is shown at the bottom (numbers in white).

DOI: https://doi.org/10.7554/eLife.35069.003

The following source data and figure supplements are available for figure 1:

**Source data 1.** Raw data for IVT sgRNA versus 2-part cr:tracrRNA-based V5 knock-in efficiency in NS and GNS cells.
DOI: https://doi.org/10.7554/eLife.35069.006
**Figure supplement 1.** PCR genotyping and Sanger sequencing of V5 knock-in bulk populations show error-free insertion of the tag-encoding sequence.
DOI: https://doi.org/10.7554/eLife.35069.004
**Figure supplement 2.** PCR genotyping and Sanger sequencing of V5 knock-in bulk populations show error-free insertion of the tag-encoding sequence.
DOI: https://doi.org/10.7554/eLife.35069.005

(ICC) for the V5 fusion protein (*Figure 1C*). The csRNA-based RNP (csRNP) gave a >4-fold and >10-fold increase in V5 knock-in efficiency for *Olig2* and *Sox2*, respectively (*Figure 1D*). Improved knock-in efficiencies were also obtained using an independent cell line (glioma-initiating neural stem cells; termed GNS) (*Figure 1D*). PCR genotyping and Sanger sequencing confirmed in-frame and error-free insertion of V5 tag sequence at the C-terminus of *Olig2* and *Sox2* loci (*Figure 1—figure supplement 1A*). V5-positive cells all displayed the anticipated nuclear localisation and levels with no indication of non-specific expression.

Knock-in efficiency might vary when using distinct biochemical tags. We therefore tested a variety of widely used alternative tags (V5, HA, 3XFLAG, or Myc). Each tag varied substantially in size, and consequently homology arm length (*Table 1*). Nevertheless, we observed similar rates of knock-in

**Table 1.** Larger cargos (3XFLAG) could be inserted with high knock-in frequency at mouse Sox2 locus.

**Mouse NS cells (ANS4)**

| Tag | Tag size (bp) | Homology arms (bp) | Knock-in efficiency (%) |
|---|---|---|---|
| 3XFLAG | 66 | 67 | 21.2 |
| V5 | 42 | 79 | 17.6 |
| Myc | 30 | 85 | 14.9 |
| HA | 27 | 86 | 15.8 |

Mouse NS cells (BL6)

| Tag | Tag size (bp) | Homology Arms (bp) | Knock-in efficiency (%) |
|---|---|---|---|
| 3XFLAG | 66 | 67 | 13.7 |
| V5 | 42 | 79 | 13.9 |
| Myc | 30 | 85 | 9 |
| HA | 27 | 86 | 14.6 |

DOI: https://doi.org/10.7554/eLife.35069.007
The following source data available for Table 1:
**Source data 1.** Raw data for epitope tag knock-in efficiency using 3XFLAG, HA, V5, and Myc single-stranded donor DNA templates.
DOI: https://doi.org/10.7554/eLife.35069.008

(15–21%) across all four tags for *Sox2* (*Figure 1E*; *Table 1*). An independent adult NS cell culture also gave similar results (9–15% knock-in efficiency, *Figure 1—figure supplement 1B*).

High knock-in efficiencies were not limited to *Sox2* and *Olig2*. We found that *Sox3*, *Sox8*, and *Sox9* – three *Sox* family members that are expressed in NS cells – had knock-in efficiencies of 30%, 14% and 26%, respectively (*Figure 1F*; *Figure 1—figure supplement 1C* and *Figure 1—figure supplement 2A*). Furthermore, we could simultaneously knock-in two different tags (*Sox2*: HA tag; and *Olig2*: V5 tag) in the same cells using a single transfection (4% double-positive cells, *Figure 1—figure supplement 2B*).

Altogether, these results indicate that use of the modified two-part synthetic cr/tracrRNA system is more effective than IVT for epitope knock-in in mammalian NS and GNS cells. Using csRNP delivery, we achieved 5–30% knock-in efficiency across distinct cell lines for different genes and using different tags. Notably, this was accomplished without the requirement for flow cytometry or plasmid-based selection strategies.

## The high frequency of knock-in using csRNP facilitates simple recovery of biallelic-tagged clonal lines

Generation of biallelic-tagged clonal lines could be advantageous for downstream applications, as all target protein will be tagged, enabling improved signal-to-noise ratios in assays. Low efficiency of tagging requires screening of hundreds or thousands of clones that need to be screened and geno-typed, limiting downstream applications. Heterozygous lines might also harbour indels on the other non-HR untagged allele (*Merkle et al., 2015*; *Bressan et al., 2017*), which may cause inappropriate transcriptional/post-transcriptional regulation.

The improved knock-in efficiency using the csRNP method encouraged us that recovery of bial-lelic clonal lines might be straightforward. Clonal NS cell lines were established from bulk tagged populations following single-cell deposition to 96-well plates. Tagged clones were then identified following replica plating and ICC for V5 tag (*Figure 2A*). Biallelic clones were scored using PCR primer-pairs flanking the tag sequence (*Figure 2B*) and validated using ICC and Sanger sequencing (*Figure 2C*; *Figure 2—figure supplement 1*).

Eighty-nine clonal lines were generated from the *Sox2*-V5 knock-in cells. Thirteen of these were V5-positive by ICC and 11 were correctly targeted as confirmed by PCR (*Table 2*). Of these four had successfully integrated the V5 tag sequence at both alleles (30% of V5-positive clones) (*Table 2*). High frequencies of bi-allelic knock-in were also obtained for *Sox3*-V5 (62.5%). We also derived several biallelic knock-in lines from another independent cell line (IENS, mouse GNS cells): *Sox2*, *Sox3*, *Sox8* and *Sox9* (7%, 26%, 57%, and 15% biallelic knock-in of V5-positive clones, respectively) (*Table 2*). Thus, biallelic-tagged clonal lines can be readily recovered.

## Multiple stem cell-types can be epitope-tagged using csRNPs, including non-expressed genes

To test the general applicability of the csRNP tagging method across other types of stem cells, we compared head-to-head tagging efficiencies between mouse ES cells and NS cells. We initially focussed on four transcription factors (TFs): *Sox2, Sox3, Ctcf* and *Pou3f1*, and the chromatin regula-tor *Ezh2*; each of these is expressed in both cell types. In each case, we found that mouse ES cells (E14Tg2a) were tagged at similar level of efficiency to the NS cells (knock-in efficiency range 6–11%, *Figure 3A,B*).

Non-expressed genes are often difficult to engineer. We therefore tested csRNPs for several neu-ral-affiliated TFs (*Sox9, Pou3f2,* and *Pou3f3*) that are expressed in NS cells but not ES cells (*Figure 3C*). V5 insertion was first confirmed by PCR genotyping in the bulk populations and sug-gested both ES cells and NS cells were effectively tagged (*Figure 3—figure supplement 1*). V5-tagged protein was detected by ICC only in NS cells and not in ES cells (*Figure 3C*, left and middle panels). However, for each of these genes, upon neural lineage differentiation of the ES cells, we noted a proportion of the Nestin-expressing neural rosettes that were V5-positive; 10.5%, 20%, and 11.8% for Sox9, Pou3f2, and Pou3f3, respectively (*Figure 3C*, right panel). Thus, non-expressed genes can be successfully tagged in ES cells, without deploying any selection strategies or plasmids. We also assessed csRNP-based tagging in human ES cells. V5 knock-in was successfully

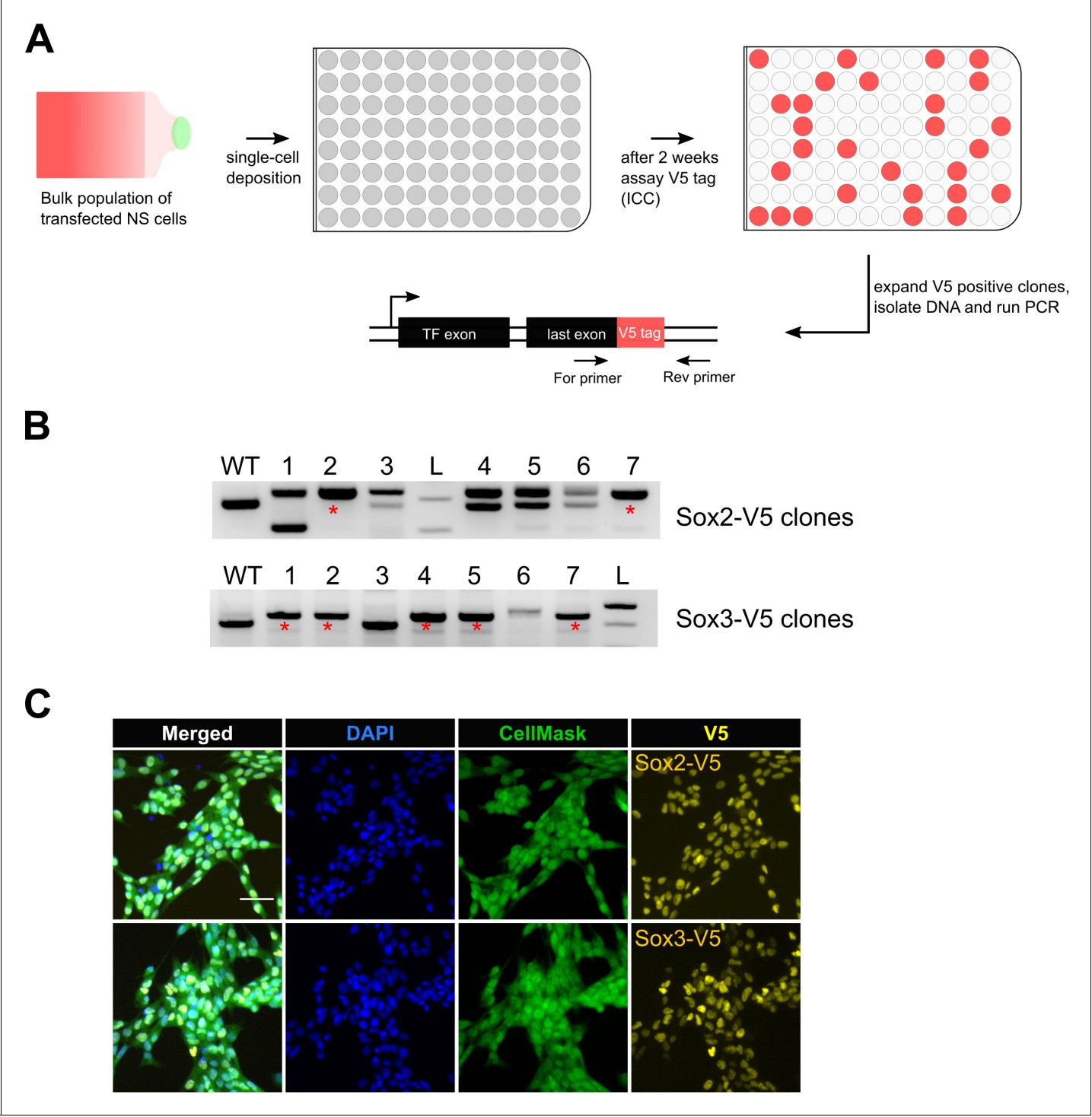

**Figure 2.** Bi-allelic knock-in clonal lines can be readily generated. (**A**) Experimental workflow for generating clonal lines. Cells from the bulk population cultures were single-cell deposited to 96-well plates using flow cytometry. Two weeks later, V5-positive clones were selected by ICC assay and confirmed by PCR genotyping. (**B**) Representative agarose gel pictures for *Sox2-V5* and *Sox3-V5* PCR genotyping are shown. Wild-type unedited cells (control) are marked with WT; 1 kb+ DNA ladder is marked with L; bi-allelic clones are marked with red asterisks. (**C**) ICC images of bi-allelic V5 knock-in clones are shown for Sox2 and Sox3 TFs. Nuclear stain DAPI (blue), entire cell stain CellMask HCS (green), and V5 tag (yellow) are shown.
DOI: https://doi.org/10.7554/eLife.35069.009

The following figure supplement is available for figure 2:

**Figure supplement 1.** Sanger sequencing of V5 knock-in clonal lines.

*Figure 2 continued on next page*

*Figure 2 continued*

DOI: https://doi.org/10.7554/eLife.35069.010

demonstrated in human ES cells for *SOX2* (*Figure 3D*). We also tested human GBM-derived cells for *OLIG2*, *SOX2*, and *SOX9* genes (15–70% efficiency) (*Figure 3D*).

These data illustrate the power of the csRNP-mediated efficient tagging of non-expressed genes in stem cells, and subsequent monitoring of the tagged protein in their differentiating progeny. We also conclude that the same csRNP epitope tagging approach and reagents can work effectively across diverse mouse and human pluripotent stem cells, neural stem cells, and cancer-derived stem cells.

## DSB distance from the stop codon is a key parameter for successful tagging

To further define the parameters influencing the reliability and efficiency of tagging, we attempted V5 epitope knock-in at the C-terminus for all 50 *Sox* and *Fox* genes. This set of genes included both expressed and non-expressed genes. Previous studies have indicated that the distance of the DSB to the insertion site influences the frequency of successful HR (*Bialk et al., 2015*; *Paquet et al., 2016*; *Liang et al., 2017*). We designed two different target crRNAs in the 3'UTR of the gene for each of the 50 target genes; one cutting proximal and the other distal to the stop codon. For each of the TF, cells were transfected with Cas9 RNP containing either of the crRNA and a common matched ssODN to assess if distance of the cut site from the stop codon influenced knock-in efficiencies (*Figure 4A*).

By PCR genotyping, we found that both proximal and distal gRNAs could result in successful tagging in the majority of cases; 30/50 genes (60%) for proximal DSB and 27/50 for the distal DSB (54%) (*Figure 4—figure supplements 1, 2* and *3*). However, this assay is qualitative. To quantitatively score the knock-in efficiency, we performed V5 ICC assay for the seven expressed TFs (*Figure 4B,C*). Sanger sequencing confirmed targeted insertion of the V5 tag-coding sequence (*Figure 4D*). Importantly, by comparing the efficiency of tagging for these seven genes, we noted a consistent trend towards increased tagging efficiency for the most proximal cut site. For example, Sox3 showed 18% and 5% knock-in efficiency, for proximal and distal gRNAs, respectively (*Figure 4B*). For four genes (*Sox9, Foxj3, Fok1* and *Foxk2*), the distal gRNA did not work, whereas the proximal gRNA facilitated high knock-in efficiency (*Figure 4C*). There were no genes for which the more distal gRNA worked better than the proximal gRNA. These results suggest proximity of DSB to the stop codon influences the efficiency of knock-in.

**Table 2.** Summary of knock-in lines generated using csRNP delivery.

Table summarising knock-in lines derived from mouse NS and GBM-model cells. The percentage bi-allelic knock-in among all V5-positive clones, as confirmed by PCR genotyping, is listed in the last column.

| Cell type | Gene | Colonies picked | V5 positive (ICC) | Correctly targeted (PCR) | Bi-allelic clones by PCR (% of total V5 positive) |
|---|---|---|---|---|---|
| Mouse NS cells | *Sox2* | 89 | 13 | 11 | 4 (30.8%) |
| | *Sox3* | 55 | 8 | 6 | 5 (62.5%) |
| Mouse GBM-model cells | *Sox2* | 242 | 14 | 14 | 1 (7.1%) |
| | *Sox3* | 96 | 19 | 18 | 5 (26.3%) |
| | *Sox8* | 96 | 7 | 6 | 4 (57.1%) |
| | *Sox9* | 469 | 46 | 41 | 7 (15.2%) |
| | *Foxk2* | 96 | 2 | 2 | 1 (50%) |

DOI: https://doi.org/10.7554/eLife.35069.011

The following source data available for Table 2:

Source data 1. Raw data for V5 ICC and PCR genotyping of clonal knock-in lines.

DOI: https://doi.org/10.7554/eLife.35069.012

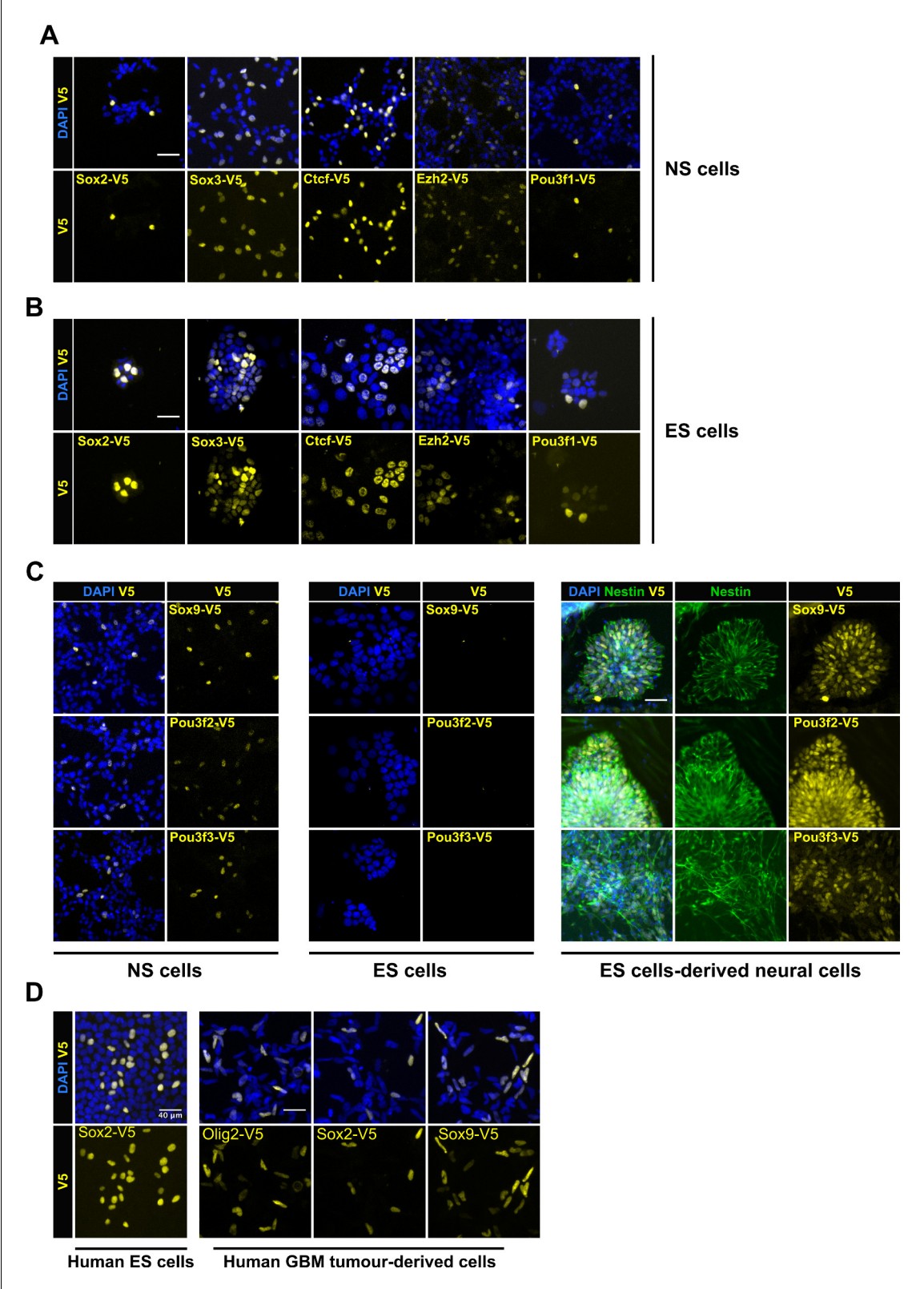

**Figure 3.** | Variety of cell types can be epitope-tagged using RNP method. Representative V5 ICC images are shown for V5 tagging of five TFs that are expressed in both mouse NS cells (**A**) and ES cells (**B**). V5 ICC images for the three neural-specific TFs Sox9, Pou3f2, and Pou3f3 in mouse NS cells (Figure **C**, left panel). Mouse ES cells were electroporated with similar three TF-reagents (Figure **C**, middle panel). Later, ES bulk populations of non-expressed TFs were differentiated into neural lineage and assayed for expression of TF-V5 fusion proteins by ICC (Figure **C**, right panel). Differentiation

*Figure 3 continued on next page*

*Figure 3 continued*
into neural stem progenitors was confirmed by Nestin ICC (in green), only V5-positive rossettes for each of the three genes are shown. (**D**) V5 ICC images for epitope tagging using human ES cells and GBM patient-derived cell lines.
DOI: https://doi.org/10.7554/eLife.35069.013
The following figure supplement is available for figure 3:

**Figure supplement 1.** PCR genotyping and sequence confirmation of V5 tagging in mouse ES cells.
DOI: https://doi.org/10.7554/eLife.35069.014

## A scalable pipeline for high-throughput knock-in of epitope tags using 96-well microplates

It is often desirable to explore large numbers of proteins within a shared family, complex or pathway. Methods enabling knock-in of many genes in parallel would be valuable. The gene-specific synthetic short crRNAs and matched ssODN repair templates can be obtained from commercial suppliers in 96-well microplates. Indeed, all steps can be performed easily in 96-well format: preparation of the transfection-ready components via automated liquid handling, benchtop incubation/annealing, 96-well transfection, and automated microscopy to acquire images across 96-well plates. We reasoned that scale-up could therefore be relatively straightforward. A major remaining bottleneck, however, is the need for bioinformatics design tools specifically tailored towards epitope tagging applications.

Manually extracting gene sequence data, identifying appropriate gRNAs, and design of modified repair ssODNs, would be laborious and error-prone for hundreds of genes. To automate the batch design of crRNAs and ssODNs, we developed 'Tag-IN', a novel web-based tool (*Figure 5*, http://tagin.stembio.org). This enables design of appropriate gRNA and ssODNs for both human and mouse species, with flexibility in choice of tag. Our 'Tag-IN' incorporates key design rule and parameters – e.g. incorporating 'Rule Set 2' for maximised activity (*Doench et al., 2016*), and 'MIT' scoring model (*Hsu et al., 2013*) to minimise off-target effects. Our tool also considers optimal distance from the insertion site (stop codon), and outputs the matched ssODNs modified with PAM-blocking mutations and appropriate epitope tag sequence. 'Tag-IN' also enables batch design; critical for the effective scale up to 96-well format (*Figure 5*).

Using the Tag-IN tool, we designed crRNAs and matched V5 encoding repair ssODNs against 185 different transcription factors. Each of these crRNAs/ssODNs were also manually verified to ascertain that the tool picked proximal-cutting crRNA and the ssODNs contained appropriate PAM-blocking mutations and homology arms. These mouse genes were selected based on expression in human glioblastoma stem cells. One gRNA was tested for each gene. The RNPs were prepared using a 96-well head liquid handling device (Felix, CyBio) and then transfected in parallel into mouse GNS cells (*Figure 5*). Five days later, ICC was performed and V5 knock-in efficiencies were quantified across the entire plate using automated plate image capture (Operetta high-content imaging system, Perkin Elmer). *Sox2* tagging was used as a positive control in six wells; these gave consistent V5 knock-in efficiency across the plate, confirming no 'edge-effects' during the procedure (10.5 ± 2.5%) (*Figure 6A*).

Remarkably, for the first 96-well plate, 30 out of 90 TFs were positive for V5 ICC with typical knock-in efficiencies ranging from 6% to 29% (*Figure 6A,B*). A second 96-well plate with distinct TFs showed similar knock-in efficiency (5%–38%), with 31 out of 95 TFs positive for V5 ICC (*Figure 6C, D*). These are similar efficiencies to those observed in our earlier single transfections (*Figures 1* and *2*). V5 ICC confirmed the expected nuclear localisation of these TFs (*Figure 6B*). Thus, ~30% of genes were successfully tagged at the first attempt with good knock-in efficiency (*Figure 6E*).

Interestingly, two of the V5-positive TFs- *Cbfb* and *Ybx1*, showed nucleocytoplasmic localisation (*Figure 6B,D*). This illustrates the valuable information regarding protein localisation and levels data that can be quickly obtained. The frequency of successful genes tagged as scored by V5 ICC in these experiments is likely to be an underestimate, as ~20% of the failed TFs are low or non-expressed NS lines (Pollard lab, unpublished data). These results clearly demonstrate the ease with which our method can be scaled to 96-well format epitope tagging.

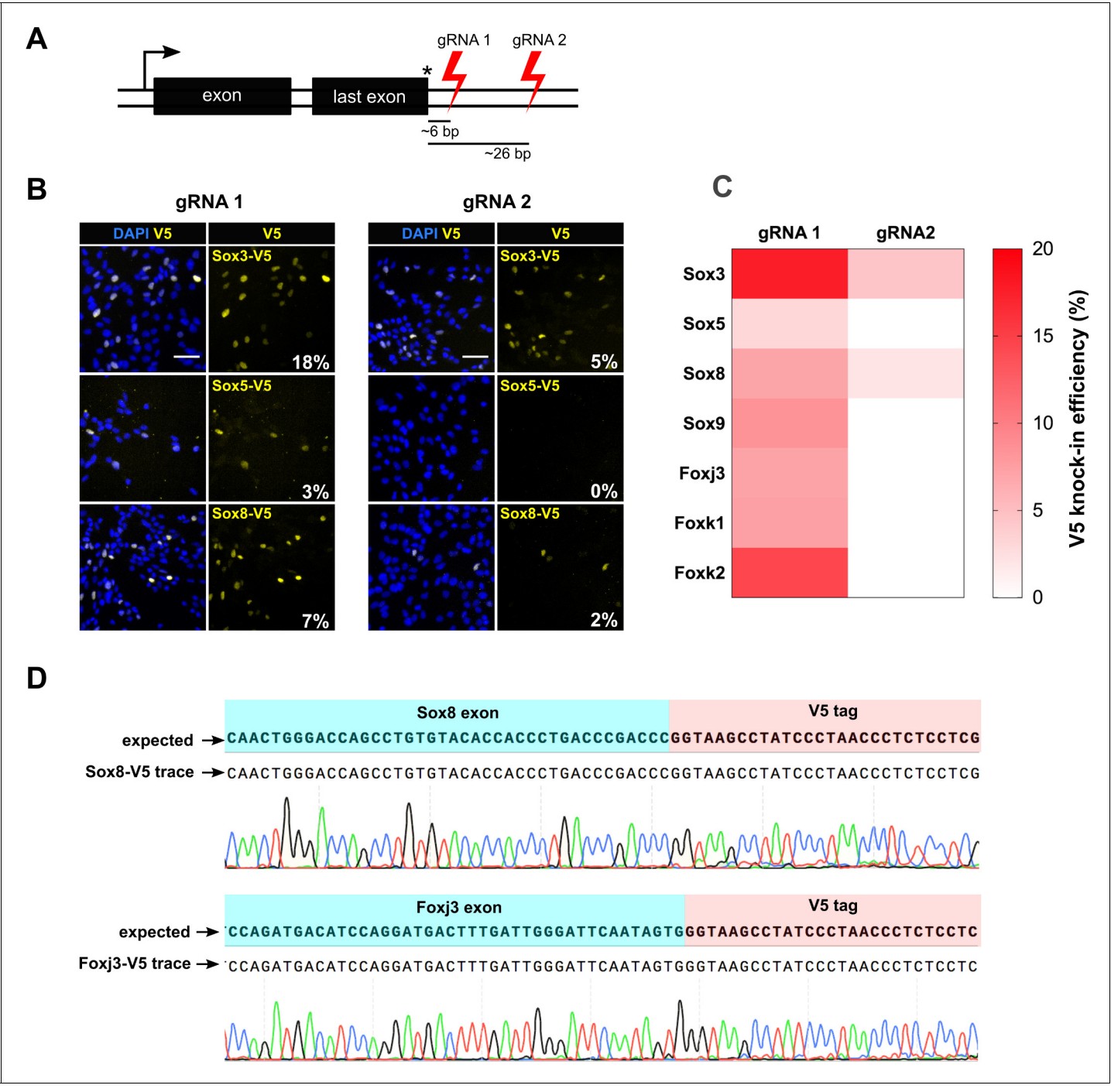

**Figure 4.** csRNP cutting proximal to insertion site is more efficient and allows knock-in at non-expressed genes. (**A**) Schematic showing the relative position of guide RNAs in the 3' UTR. The first set of gRNA (gRNA1) cuts proximal to the stop codon, the second set (gRNA 2) cuts distal. Average distance of cut site from the insertion site is shown for both sets of gRNAs against 50 TFs. (**B**) Representative V5 ICC images of Sox3, Sox5, and Sox8 V5 knock-in bulk populations obtained with gRNA 1 (left panels) and gRNA 2 (right panels) are shown; % V5 knock-in efficiency for each experiment is shown at the right bottom of V5 panels (numbers in white). (**C**) Heatmap representation of V5 knock-in efficiency obtained with gRNA1 and gRNA2 for different genes, color code is shown on the right. (**D**) Example of sequencing traces using the bulk populations of V5-tagged cells for *Sox8* and *Foxj3*. Alignment with the expected TF-V5 chimeric sequence is shown.

DOI: https://doi.org/10.7554/eLife.35069.015

The following source data and figure supplements are available for figure 4:

**Source data 1.** Raw data for V5 knock-in efficiency using two sets of guide RNAs.

DOI: https://doi.org/10.7554/eLife.35069.019

*Figure 4 continued on next page*

*Figure 4 continued*

**Figure supplement 1.** V5 tag-specific PCR amplification of bulk populations gene-edited using first set of gRNAs.

DOI: https://doi.org/10.7554/eLife.35069.016

**Figure supplement 2.** V5 tag-specific PCR amplification of bulk populations gene-edited using second set of gRNAs.

DOI: https://doi.org/10.7554/eLife.35069.017

**Figure supplement 3.** Sanger sequencing confirms error-free insertion of V5-encoding sequence at the C-terminus of expressed and non-expressed genes.

DOI: https://doi.org/10.7554/eLife.35069.018

## csRNP-derived knock-in lines can be used for immunoprecipitation-mass spectrometry (IP/MS) to identify protein partners

As a proof-of-principle of the applications, we performed V5-immunoprecipitation followed by mass spectrometry (IP-MS), to identify interaction partners of *Olig2* in mouse GNS cells using the RIME (*Mohammed et al., 2016*) and ChIP-SICAP (*Rafiee et al., 2016*) methods. These enable identification of chromatin-bound protein partners – the latter being more stringent for chromatin-bound proteins. For each assay, we used V5 monoclonal antibody conjugated to magnetic beads. RIME analysis showed high enrichment of the bait protein Olig2 (*Figure 7A*) and subunits of SWI/SNF complexes and histone deacetylases (HDACs) in the pull-down complexes (*Figure 7—source data 1*). Physical interaction of Olig2 and SWI/SNF complex has been previously reported and this interaction is essential for oligodendrocyte differentiation (*Yu et al., 2013*). HDACs have a known functional role in Olig2 function during development.

ChIP-SICAP analysis showed strong enrichment of Olig2 and core histone proteins suggesting specific pull-down of chromatin fragments bound by Olig2 TF (*Figure 7B*). Noteworthy, two other oligodendrocyte lineage transcription factors, Olig1 and Olig3, were detected among the most enriched proteins. Earlier studies reported that sets of genes regulated by various Olig proteins have a partial overlap (*Ligon et al., 2007*; *Meijer et al., 2012*), explaining co-occupation of the same DNA-sites as shown by these data. Furthermore, we detected two other members of the basic helix-loop-helix (bHLH) family: Npas3 and Tcf4. The bHLH transcription factors are known to form heterodimers with other bHLH proteins on chromatin (*Massari and Murre, 2000*) and the presence of Npas3 and Tcf4 may be explained by their direct physical interaction (*Figure 7B*; *Figure 7—source data 1*). Both Npas3 and Tcf4 have been reported to be involved in CNS development (*Shin and Kim, 2013*; *Chen et al., 2016*). Interestingly, a recent study reporting interactions of Olig2, Tcf4 and Npas3 in mouse neural stem cells by FLAG-affinity purification (*Moen et al., 2017*) allowed us to correlate interactomes of all the three bHLH proteins with our data. Taken together, our results indicate that Tcf4 and Npas3 co-localise with Olig2 on chromatin, suggesting a functional interaction.

In addition to protein interactions of Olig2 on-chromatin, we also analysed the flow-through of the streptavidin enrichment, representing interactions with soluble Olig2 (*Supplementary file 3*). As expected, Olig2, Olig1, and Olig3 were found among the most highly abundant proteins, indicating high specificity of the immunoprecipitation. Furthermore, composition of identified proteins correlates with previous findings: we detected three known interactors of Olig2 (Cul3, Smarca4 and Sox8 (BioGRID, Intact)) and 55 other proteins reported to co-precipitate with Olig2 (*Moen et al., 2017*) (*Supplementary file 3*). These included many SWI/SNF family members, and chromatin regulators Cbx3 and Chd4, consistent with RIME. Collectively, our data confirm that V5-tagging can be effectively combined with ChIP-SICAP to identify proteins that co-localise on chromatin or that interact off-chromatin.

## Knock-in of mCherry fluorescent reporter in NS cells using dsDNA blocks and csRNP

Insertion of fluorescent protein-encoding sequences (such as GFP, mCherry) in frame within gene coding regions enables monitoring of eukaryotic protein localisation in live cells. Additionally, fusion proteins can be used for pull-down assays (immunoprecipitation and ChIP). Encouraged by the facile deployment and efficiencies of the Cas9 RNP method for small epitope tagging, we next asked if the same approach could be used for knock-in of relatively large mCherry-encoding DNA sequence

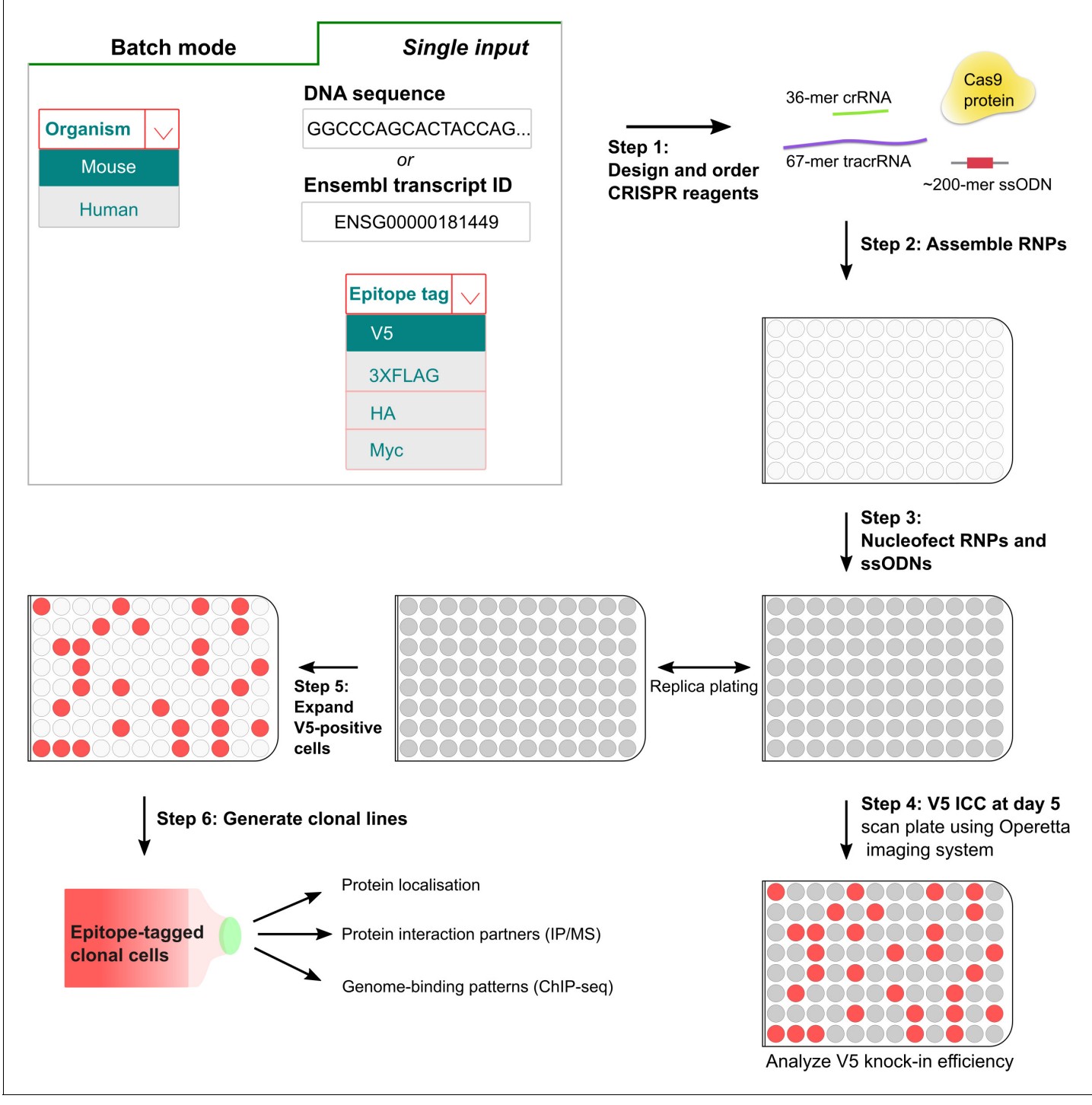

**Figure 5.** A simple pipeline for high-throughput epitope knock-in using 96-well format. In the first step, crRNAs and matched ssODNs are designed (either single input or batch processing) using the 'Tag-IN' bioinformatics tool. The tool picks top two crRNAs in the 3'UTR (within 8–15 bp from stop codon if high-quality crRNAs available) and designs an ssODN for each query based on user's choice of the epitope tag. After the procurement of CRISPR ingredients from the supplier, the RNPs and matched ssODNs are assembled in vitro in 96-well plates (*step 2*) and transfected into stem cells using Amaxa shuttle system (*step 3*). Five days after the transfection, a replica plate is processed for V5 ICC and images are captured using high-content imaging system Operetta (*step 4*). V5-positive cells from the corresponding wells can be later expanded to derive clonal lines for downstream applications (*step 5,6*).

DOI: https://doi.org/10.7554/eLife.35069.020

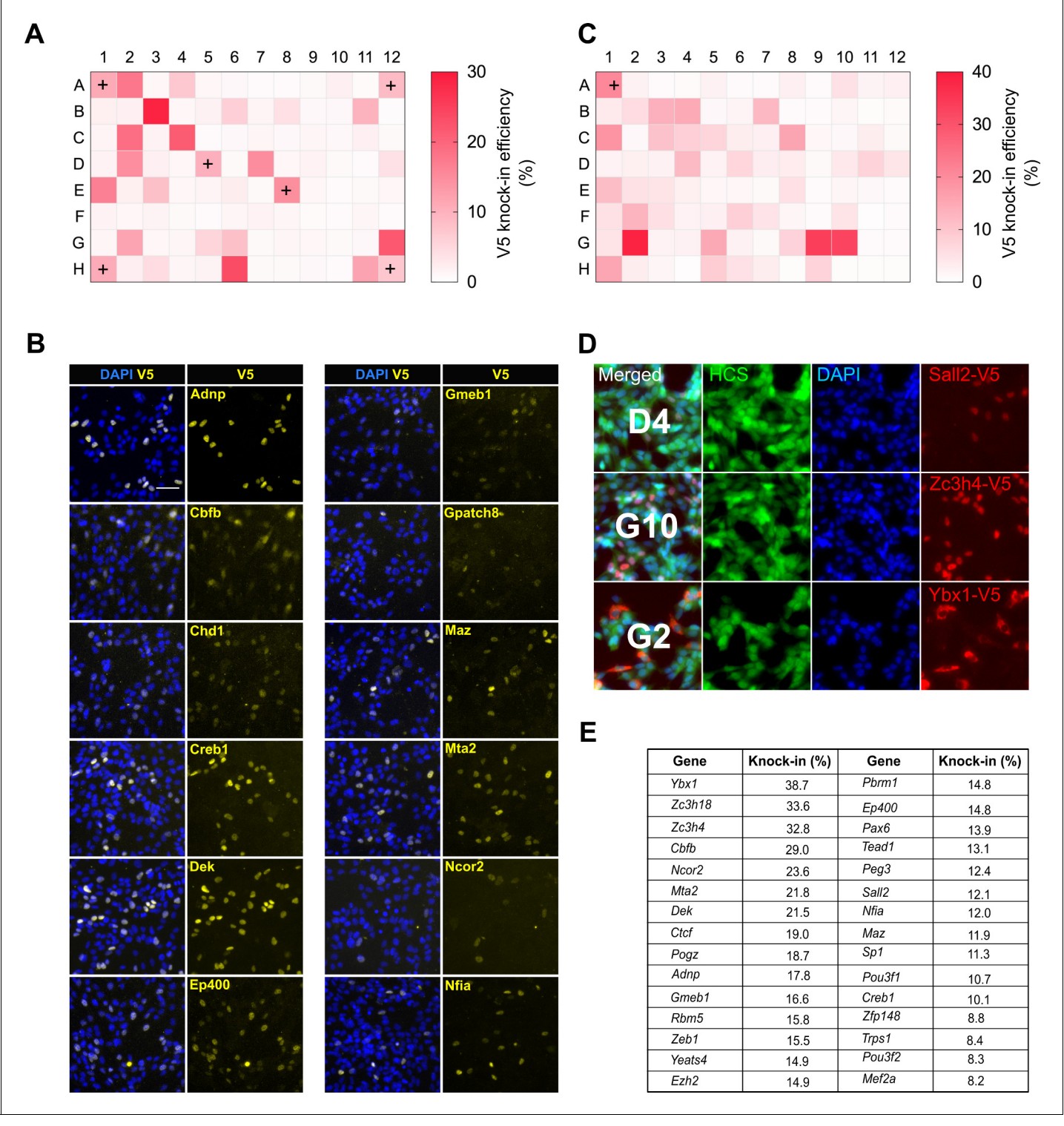

**Figure 6.** High-throughput epitope knock-in in a 96-well format. (**A**) Heat map of V5 knock-in efficiency across the entire 96-well plate. The positive control Sox2 was included in six different wells (marked with '+'). (**B**) Representative ICC images of V5 knock-in from figure (a) are shown, nuclear stain DAPI (blue) and TF-V5 fusion protein (yellow) are shown. (**C**) A 96-well plate heatmap of V5 knock-in for another set of 96 TFs. (**D**) Representative V5 knock-in panels, as obtained from the Operetta system, for the 96-well plate from (**C**) are shown, well number is indicated in the merged panel of each TF. Nuclear stain DAPI (blue), entire cell stain CellMask HCS (green), and V5 tag (red) are shown. (**E**) Top 30 TFs with highest knock-in efficiency from (**A**) and (**C**).

DOI: https://doi.org/10.7554/eLife.35069.021

*Figure 6 continued on next page*

*Figure 6 continued*

The following source data is available for figure 6:

**Source data 1.** Raw data for V5 knock-in efficiency across the 96-well plate.

DOI: https://doi.org/10.7554/eLife.35069.022

(~700 bp) in NS cells. First, we tested the effect of variable length of homology arms on knock-in efficiency (*Figure 8A*). For these experiments, double-stranded linear DNA fragments harbouring variable homology arms were used as donor DNA templates (PCR-amplified from a previously reported promoterless Sox2-mCherry plasmid) (Bressan et al.). We found that dsDNA with larger homology arms were more effective in mCherry knock-in (*Figure 8B*), although homology arms as small as 100 bp showed ~0.29% cells positive for mCherry in the bulk populations. We next tested mCherry knock-in at *Olig2* and *Foxg1* loci and achieved 0.34% and 0.11% efficiency, respectively (*Figure 8C*). Live-imaging of mCherry-sorted populations for the three genes (*Sox2*, *Olig2*, and *Foxg1*) showed the expected localisation and levels of fusion proteins in NS cells (*Figure 8D*). Together, these results show that our csRNP method can be used for knock-in of fluorescent reporters in NS cells.

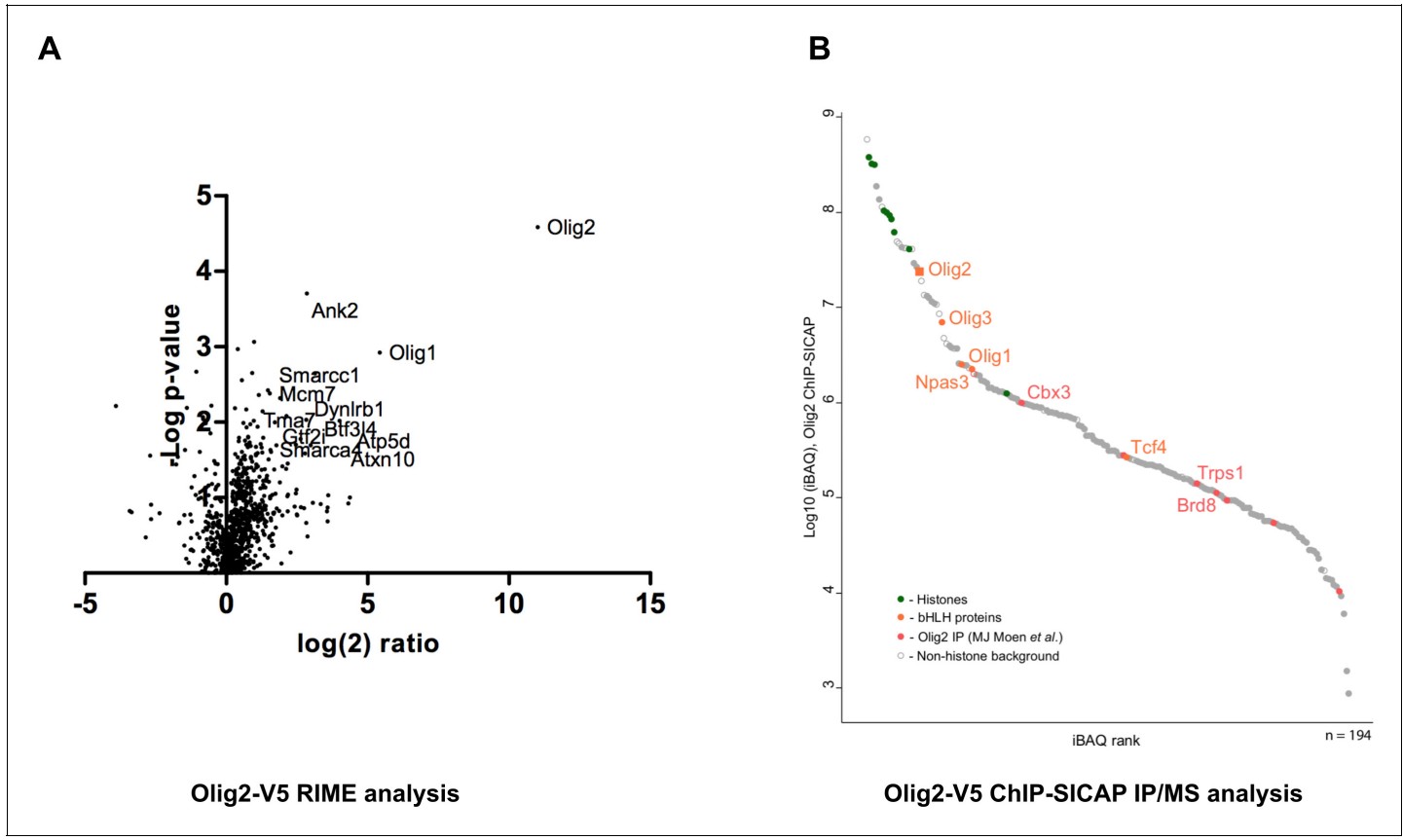

**Figure 7.** Identification of Olig2 partners using V5 knock-in lines. Mouse GNS Olig2-V5 cells were used for the identification of Olig2 interaction partners. (A) RIME analysis, volcano plot showing log(2) fold change plotted against −log(10) p value for endogenously V5-tagged Olig2 samples versus samples generated from an untagged parental cell line. (B) ChIP-SICAP analysis, proteins identified in the chromatin-bound complexes are ranked based on iBAQ score in the descending order.

DOI: https://doi.org/10.7554/eLife.35069.023

The following source data is available for figure 7:

**Source data 1.** Raw data for Olig2 interaction partners obtained from RIME and ChIP-SICAP analyses.

DOI: https://doi.org/10.7554/eLife.35069.024

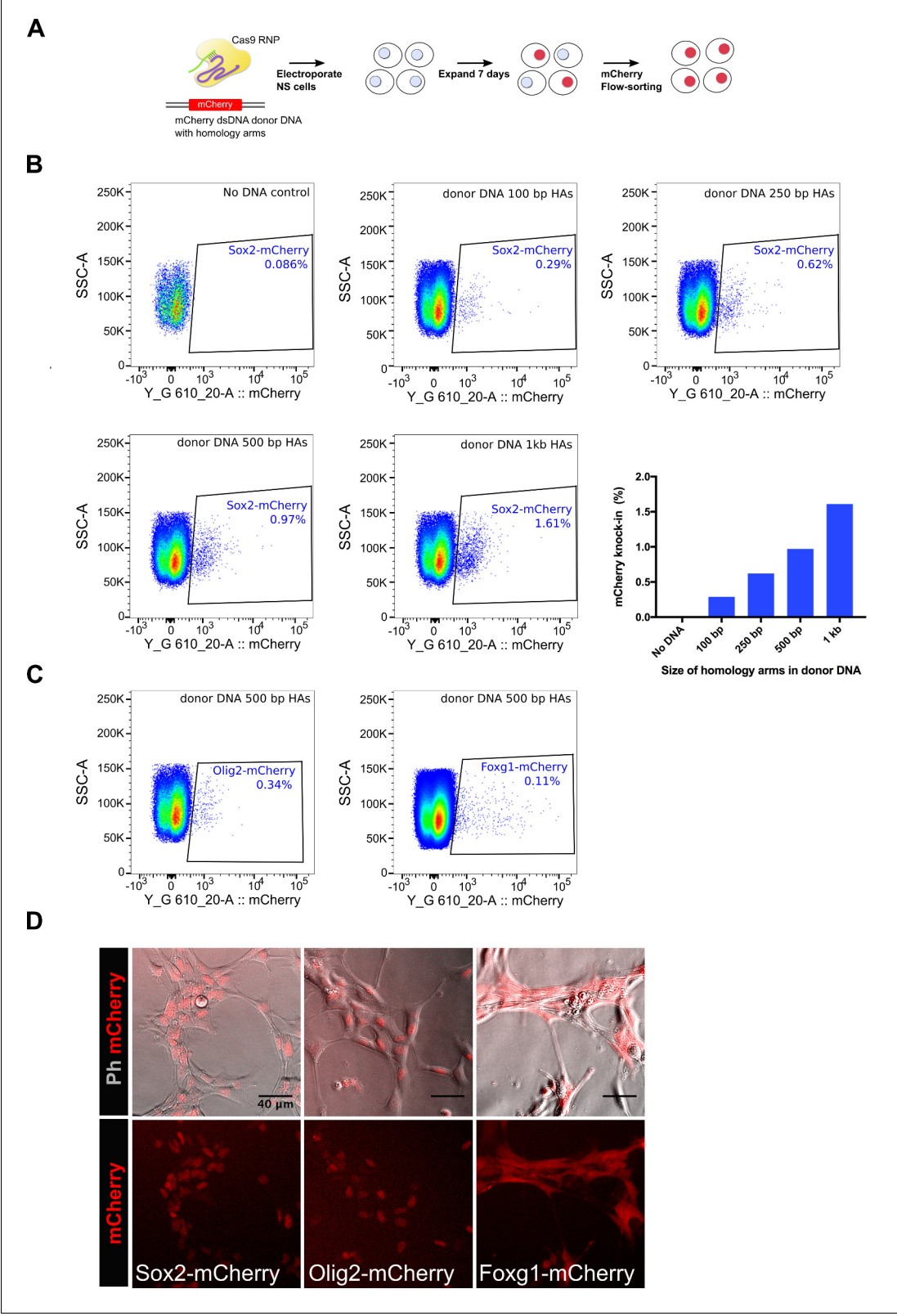

**Figure 8.** Knock-in of mCherry reporter using Cas9 RNP delivery. (**A**) Schematics of the mCherry knock-in experiment is shown. NS cells were electroplated with Cas9 RNP complex along with a dsDNA donor DNA, which was amplified using a vector plasmid that harbours mCherry encoding sequences flanked by promoter-less Sox2 homology arms. Cells were allowed to recover and expand for 7 days and then analysed by flow-cytometry for mCherry knock-in. (**B**) Flow-cytometry analysis of mCherry knock-in at Sox2 locus. Effect of variable length of homology arms was assessed,

*Figure 8 continued on next page*

*Figure 8 continued*

composite graph for all the panels is shown at the right bottom. (**C**) Flow-cytometry analysis of mCherry knock-in at Olig2 and Foxg1 loci, a donor DNA containing 500 bp homology arms was used for each gene. (**D**) Live-images of the mCherry-sorted populations are shown. Top panels show phase contrast and mCherry merged; bottom panels show mCherry alone.

DOI: https://doi.org/10.7554/eLife.35069.025

The following source data is available for figure 8:

**Source data 1.** Raw data for Sox2-mCherry knock-in efficiency using variable lengths of homology arms.

DOI: https://doi.org/10.7554/eLife.35069.026

## Discussion

An overarching goal in biology is to determine the key functions of each protein encoded in the genome. Epitope tagging of endogenous genes using CRISPR-assisted knock-in provides new opportunities to interrogate protein function, expression, subcellular localisation and interacting partners. In this study, we demonstrated simple and efficient epitope tag knock-in across a large number of genes in mouse and human stem cells.

Use of Cas9 recombinant protein is critical. We found that when combined with synthetic modified RNAs significant enhancement in the efficiency is possible. This was achieved using faster and more reliable method – compared to plasmid-based strategies. Importantly, we showed primary mammalian stem cell lines are readily amenable to RNP engineering, without the need for plasmid production, or prior genetic manipulation of the host cell lines. Our method is therefore versatile enough to be implemented by any laboratory in their existing cell lines. All reagents can now be obtained 'off the shelf'.

Assembly of RNPs with csRNAs is simple and requires only ~30 min hands-on-time; this compared favourably to IVT reactions, which require multiple steps, are time-consuming, and result in variable quality of gRNAs. In our hands, these reagents are stable at −20° C for many months, and is particularly convenient when re-used across a range of cell types. Perhaps more importantly, use of the custom synthetic cr/tracrRNAs with their modified backbone and protected ends, shields the RNA from cellular RNases. We found greatly reduced toxicity – one of the key advantages of the csRNP over IVT sgRNA and plasmids-based delivery methods.

We routinely generated clonal knock-in lines in a time-frame of 4–6 weeks. Consistent with a previous report (*Liang et al., 2015*), we found that RNP complexes target both copies of the genes at high frequency, enabling facile isolation of bi-allelic knock-in clones. Although monoallelic knock-in clones are sufficient for pull-down assays, bi-allelic clones are preferred to have more confidence in the interpretation of downstream assays.

While our focus has been on neural stem cells and their malignant counterparts (glioma stem cells), these same methods and reagents can work well in other stem cells, such as ES cells. Indeed, our knock-in data using ES cells revealed that non-expressed genes can be efficiently tagged using the same protocol. Under appropriate differentiation cues, induction of these proteins can be monitored using ICC analysis in the ES cell differentiating progeny. We believe this could be a key application – rapidly enabling assessment of proteins across a range of lineage contexts.

Despite the excitement, there are also some caveats. Foremost, inevitably there is a risk that C-terminus tagging can potentially compromise protein function, localisation or levels in the edited cells. Not all proteins will be amenable to tagging. If the protein of interest harbours critical C terminal domains, an N-terminus approach could be pursued – or knock-in to other regions such as structurally neutral linkers, if this is known. In this study, we used PAM-strand (non-complementary strand) for donor DNA synthesis, which is less likely to be cut by Cas9 complex (*Gasiunas et al., 2012*). We have not compared PAM versus non-PAM donor DNA strands for knock-in efficiency although a recent report suggests that for symmetrical PAM ssODNs, both strands are equally effective for knock-in (*Liang et al., 2017*).

Previous findings have reported a higher HDR rate with the guide RNAs that cut near the insertion site (*Bialk et al., 2015*; *Paquet et al., 2016*; *Liang et al., 2017*). Our results corroborate these findings and reveal that cut-to-insertion-site distance is a critical factor for successful knock-in. Lower knock-in efficiency achieved with the distal-cutting gRNAs could be attributed to 'partial HDR' events (*Guo et al., 2018*), which are not detectable in our ICC end-point assays. Although beyond

the scope of our present study, whole-genome sequencing of the CRISPR-edited bulk cells can shed light on the extent of 'partial HDR' events in epitope knock-in experiments. The average size of CRISPR/Cas9-induced indels in mammalian cells has been reported to be around 1–5 bp (*Paquet et al., 2016*). We therefore recommend designing guide RNAs in the 3'UTR, preferably cutting 8–15 bp downstream of stop codon, to avoid interfering with the stop codon.

The ability to scale-up tagging to 96-well format required us to tackle the bottleneck in design. Our newly developed 'Tag-IN' tool simplifies gene sequence retrieval and designing crRNA and ssODN for medium-throughput experiments and is available from any web browser (http://tagin.stembio.org). This can be modified for use with alternative design parameters and distinct tags. We demonstrated one of the key applications by single-cell imaging of the levels and localisation for 60 different transcription factors in glioma stem cells. These were successfully tagged without any screening of gRNAs or ssODN; those that failed could be due to the gRNA not working effectively or the ssODN donor being sub-optimal. These could be re-tested with replacements – particularly for the ssODN, other tag can be used that works effectively in some instances (data not shown). Alternatively, the protein maybe expressed at levels too low to detect by ICC, but the tag insertion might still be detected by PCR genotyping. This is likely the case for 10–20% of the genes we explored.

The pipeline that emerges is fully scalable and does not require sophisticated tools, expertise, or know-how. There are no significant bottlenecks in either the design, acquisition/production of reagents or delivery into cells. It was possible to generate such data with relatively little labour investment, and the whole pipeline from obtaining the reagents to imaging data was accomplished within three weeks.

Increasing the throughput further, we can envision systematic surveys of many hundreds or thousands of genes. Because tag-specific antibodies work universally across different cell types and species, this approach should allow cross-species comparisons and will also complement existing efforts to develop and characterise native antibodies. Furthermore, tagging of DNA-binding proteins will allow us to generate comprehensive genome-wide binding sites via ChIP-Seq for key transcription factors, chromatin-modifying enzymes, and other gene regulatory proteins. Although not a focus of the current study, we also find that csRNP is highly effective for gene knockout studies (Pollard laboratory, unpublished and (*Jacobi et al., 2017*).

We demonstrated proof-of-principle in identification of Olig2 protein partners. This is likely a key application for future studies; simplifying and scaling up the ability to probe protein complexes in a range of cell types and cell states. Using the Olig2-V5 knockin lines we demonstrated on-chromatin partners in mouse glioma cells using optimised RIME and ChIP-SICAP methods, validating previously reported protein interaction partners. These optimised V5 RIME and ChIP-SICAP methods can now be deployed using the same optimised protocol for other mammalian proteins, particular for those with no good quality antibodies – a key advantage of epitope tagging.

With the remarkable developments in single-cell mRNA profiling, and plans for systematic RNA and protein atlases of mammalian cell types (*Rozenblatt-Rosen et al., 2017*), there is a greater need than ever to relieve the bottleneck of exploring protein products. Descriptive maps of cell types, while valuable, must be complemented by careful and detail molecular and cellular functional studies. Our findings suggest that epitope using CRISPR/Cas-assisted knock-in is now simple and efficient enough that systematic annotation of many hundreds or thousands of endogenous will be possible in mouse and human stem and progenitor cells.

## Materials and methods

**Key resources table**

| Reagent type (species) or resource | Designation | Source or reference | Identifiers | Additional information |
|---|---|---|---|---|
| Gene (*Homo sapiens*) | SOX2 | NA | Transcript ID_ENSEMBL: ENST00000325404 | |
| Gene (*Homo sapiens*) | SOX9 | NA | Transcript ID_ENSEMBL: ENST00000245479 | |

*Continued on next page*

*Continued*

| Reagent type (species) or resource | Designation | Source or reference | Identifiers | Additional information |
|---|---|---|---|---|
| Gene (*Homo sapiens*) | *OLIG2* | NA | Transcript ID_ENSEMBL: ENST00000333337 | |
| Cell line (*Mus musculus*) | ANS4 | PMID: 28096221 | | mouse neural stem cells |
| Cell line (*Mus musculus*) | BL6 | PMID: 28096221 | | mouse neural stem cells |
| Cell line (*Mus musculus*) | IENS | PMID: 17936558 | | mouse glioma-initiating stem cells |
| Cell line (*Mus musculus*) | ES (Tg2a) | This paper | | mouse embryonic stem cells |
| Cell line (*Homo sapiens*) | MasterShef7 (MS7) | This paper | | human embryonic stem cells |
| Cell line (*Homo sapiens*) | G7 human tumour-derived | This paper | | Cell derived from GBM tumours |
| Antibody | V5 tag | eBioscience | TCM5 #14-6796-82; RRID:AB_10718239 | 1:1000 Overnight 4°C |
| Antibody | HA Tag | Cell Signalling | 6E2 #2367; RRID:AB_10691311 | 1:100 Overnight 4°C |
| Antibody | FLAG tag | Sigma-Aldrich | #F3165; RRID:AB_259529 | 1:2000 Overnight 4°C |
| Antibody | Myc tag | Cell Signalling | 9B11 #2276; RRID:AB_331783 | 1:4000 Overnight 4°C |
| Antibody | Alexa Fluor Plus Secondary antibody | Thermo Fisher Scientific | Alexa Fluor Plus 647; RRID:AB_141663 | 1:1000 1 hr RT |
| Recombinant DNA reagent | pET28a/Cas9-Cys | Addgene | Addgene, #53261 | |
| Peptide, recombinant protein | Cas9 protein | Addgene | Addgene, #53261 | |
| Chemical compound, drug | crRNAs | Integrated DNA Technologies, USA | CRISPR-Cas9 crRNA | |
| Chemical compound, drug | tracrRNAs | Integrated DNA Technologies, USA | CRISPR-Cas9 tracrRNA | |
| Software, algorithm | Tag-IN tool | This paper | http://tagin.stembio.org | for crRNA ssODN design |
| Software, algorithm | DESKGEN tool | https://www.deskgen.com/landing/cloud.html | https://www.deskgen.com/landing/cloud.html | for crRNA ssODN design |

## Cell culture

Mouse and human NS and GNS cell lines were cultured essentially as described previously (*Conti et al., 2005*; *Pollard et al., 2006*). Laminin was purchased from Cultrex, R and D Systems. ANS4 and BL6 NS cell lines have been described previously (*Bressan et al., 2017*). Mouse ES (Tg2a) cells were cultured in GMEM supplemented with 10% fetal calf serum, 1x non-essential amino acids,1x glutamine/sodium pyruvate, 1xLIF, 1x Pen/strep and 100 µM of ß-mercaptoethanol. Media was changed every day and cells passaged approximately every other day onto plates pre-coated with 0.1% gelatin. Differentiation of ES cell was performed as described previously (*Pollard et al., 2006*) with $1 \times 10^4$ cells per cm$^2$ being seeded in N2B27 complete media for 7 days, with media being changed every 1–2 days.

MasterShef7 (MS7) human embryonic stem cells (hESCs) were re-cultured in Essential 8 (E8) medium (Gibco, A1517001) on tissue culture plastic coated with Human Recombinant Laminin-521 (BioLamina, LN521) at 5 µg/ml. Routine passaging was performed by incubating cells for 5 min at 37°C in 0.5 mM EDTA in PBS. Single-cell dissociation prior to nucleofection was performed by incubating cells for 10 min at 37°C in accutase. Y-27632 (Cell Guidance Systems, SM02) was included in the culture medium at 10 µM following initial thawing and after plating following nucleofection. New engineered cell reporters described here are available upon request. No standard cell lines were used. We used primary stem cell lines. Human ES cells were provided by the UK stem cell bank and had appropriate contamination testing and authentication.

## Colony picking and clonal lines generation

Clonal cell lines were derived from the bulk populations using either single-cell deposition to 96-well plates or by manual colony picking. Single cells were deposited into 96-well plates using BD FACSAria II cell sorter. Depending on the cell lines, we obtained 30–40 colonies per 96-well plate in 2 weeks. For manual colony picking, mouse cells were seeded at clonal density (400 cells per 10 cm dish for NS cells, 100 cells per dish for GNS cells) to a 10 cm dish and incubated in the complete media for 10–12 days for colony formation. From each dish, we picked 25–30 manually with a P20 pipette. Colonies from both methods were later replica plated into 96-well plates and analysed for successful knock-in using immunocytochemistry against the V5 tag. The V5-positive clones were further expanded for DNA extraction (PCR genotyping) and cryostorage.

## Design of guide RNAs and ssODN repair templates

For manual design: the 3'UTR sequence and 500 bp sequence upstream of the 3'UTR were retrieved using Biomart tool. The final coding exon and 3'UTR features were manually annotated using SnapGene and the ~200 bp around the stop codon were used as an input for guide RNA designing. We designed guide RNAs using either the web-based tool form Desktop Genetics (https://www.desk-gen.com/landing/) or our own bioinformatics 'Tag-IN' tool (below). High scoring guide RNAs were picked for synthesis (i.e those with cut sites in the 3'UTR, preferably within 8–15 bp distance from the stop codon and minimal predicted off-target cleavage). For ssODN design, first the PAM-blocking mutations (NGG >NGC or NGT) were introduced into the SnapGene sequence and then the epitope-tag coding sequence was inserted before the stop codon. The <200 mer ssODN ultramer was chosen to be the same strand as the guide RNA (also referred to as the PAM-strand, non-complementary strand or non-targeting strand) and is comprised of: a 5' homology arm (~70 mer), the epitope tag coding sequence, stop codon, and a 3' homology arm with the PAM-blocking mutations (~70 mer). For some of the ultramers the PAM-strand synthesis had failed and, therefore, the complementary strand (non-PAM strand) was synthesised as a donor DNA.

## Custom synthetic crRNA, tracrRNA, and ssODN

Custom synthetic crRNAs, tracrRNA, and ssODNs were manufactured by Integrated DNA Technologies, USA. The RNA backbone and ends were chemically modified for protection against cellular RNases. The 36-mer crRNA contains a variable gene-specific 20-nt target sequence followed by 16-nt sequence that base-pairs with the tracrRNA. The 67-mer tracrRNA contains the gRNA-scaffold sequence as well as 16-nt sequence complementary to crRNA. The lyophilised crRNA and tracrRNA pellets were resuspended in Duplex buffer (IDT) at 100 µM concentration and stored in small aliquots at −80°C. ssODN donor DNAs lyophilised pellets were supplied without modifications and resuspended in Duplex buffer (IDT) at 30 µM concentration.

## Production of in vitro-transcribed sgRNA

DNA template for T7-driven synthesis was prepared by annealing 119-mer, single-stranded, complementary ultramers (from IDT) encoding T7 promoter, guide RNA, and gRNA scaffold sequences. 200 ng of the template were used to synthesise sgRNA with the MEGAscript T7 Transcription Kit. The sgRNA was further purified using MEGAclear Transcription Clean-Up Kit and stored at −80°C.

## Production of recombinant Cas9 protein

BL21(DE3) cells (New England Biolabs, C2527) were transformed with the plasmid pET28a/Cas9-Cys (Addgene, Cambridge, USA, plasmid #53261) using standard protocols. Cas9 protein expression was induced with 0.5 mM IPTG (Isopropyl β-D-1-thiogalactopyranoside) (Fisher, 10715114) and the cells were incubated overnight at 20°C. 24 hr later, bacterial pellets were resuspended in 20 ml of lysis buffer (20 mM Tris-HCl pH 8.0, 500 mM NaCl, 1 mM TCEP, 5 mM imidazole pH 8.0), sonicated and loaded on a HisTrap HP 5 ml column (GE, 17-5248-01). The Cas9 protein was collected in elution buffer (20 mM Tris-HCl pH 8.0, 250 mM NaCl, 10% glycerol, 1 mM TCEP, 250 mM imidazole pH 8.0). The fractions containing Cas9 protein were pooled and loaded into a HiPrep 26/10 Desalting Column (GE, 28-4026-52) to equilibrate in Cas9 buffer (20 mM HEPES-KOH pH 7.5, 150 mM KCl, 1 mM TCEP). The purified Cas9 protein was further concentrated using Vivaspin columns (Vivaspin20, 30 000 MWCO PES, Sartorius stedim, VS2021) as per the users-guide instructions.

## Assembly of the active ribonucleoprotein (Cas9 plus csRNAs)

Synthetic Alt-R CRISPR/Cas9 crRNAs and tracrRNA were supplied by IDT. We prepared Cas9 RNP complexes immediately before electroporation experiments (a detailed protocol in a separate Appendix 1 is available). Cas9 RNPs with IVT sgRNA were assembled (1–3 µg of IVT sgRNA with 5–10 µg of Cas9 protein) as described previously (*Bressan et al., 2017*). For csRNP preparation, 100 picomoles of each crRNA and tracrRNA were annealed using gradual step-down cooling in the PCR block (5 min at 95°C, step cool-down from 95°C to 25°C at ramp rate 0.1°C/s, 4°C (store) at ramp rate 0.5°C/s). Ribonucleoprotein (RNP) complexes were assembled by adding 10 µg of recombinant Cas9 protein to the annealed cr/tracrRNAs, incubated at room temperature for 10 min and stored on ice until electroporation into cells. 30 picomoles of single-stranded donor DNA were added to RNP complexes just before electroporation to prepare the complete RNP mix. For mCherry knock-in, csRNPs were prepared similarly and 200 ng of PCR products were used as donor DNA templates per reaction. For multiplex epitope tagging, 100 picomoles of cr/tracrRNA of each Sox2 and Olig2 were mixed together with 20 µg of rCas9 protein.

## Cell transfection

We used 4D Amaxa nucleofection system for the delivery of CRISPR ingredients. For NS cells and GNS cells, approximately $1.5 \times 10^5$ cells were resuspended in 20 µL of Lonza SG cell line buffer and were mixed with the complete RNP mix and electroporated using the DN-100 program (two consecutive pulses for mouse NS cells) or using EN-138 program (one pulse for human GBM-derived cells). For embryonic stem cells, approximately $6 \times 10^4$ cells in 20 µL of Lonza P3 primary cell buffer were used for each transfection with different programs: one pulse of program CA-120 for mouse ESCs; program CB-150 for human ESCs. After the electroporation, cells were transferred into a 6-well plate and allowed to recover for 3–5 days and later seeded into 96-well plates ($1–2 \times 10^4$ cells per well) for ICC.

For scale up, RNP assembly and delivery were performed as above, except that RNP complexes were prepared a day before and stored at −20°C. Electroporation was performed using the 96-well Shuttle device (Amaxa, Lonza). Immediately after transfection cells were transferred into a 96-deep-well plate and replica plates for immunocytochemistry assay were prepared by dispensing $1 \times 10^4$ cells into 96-well plate using CyBi-FeliX Liquid Handling Platform.

## Immunocytochemistry and imaging

We performed ICC on 96-well plates 5 days after transfection. Cells were washed once with PBS and fixed using 4% paraformaldehyde for 10 min at room temperature and then permeabilised in PBST (PBS + 0.1% Triton X-100) for 20 min. Samples were incubated with blocking solution (1% goat serum in PBST) for 30 min at room temperature to block non-specific binding of the antibodies. Samples were treated overnight with primary antibodies in blocking solution followed by incubation with appropriate secondary antibodies and 4′,6-diamidino-2-phenylindole (DAPI). Images were acquired using either a Nikon wide-field fluorescence microscope or a PerkinElmer Operetta high-content imaging system. V5-positive cells were scored using Fiji software.

The following primary antibodies were used: V5 tag (eBioscience, TCM5 #14-6796-82,1:1000); HA Tag (Cell Signalling, 6E2 #2367, 1:100); FLAG tag (Sigma-Aldrich, #F3165, 1:2000); Myc tag (Cell Signalling, 9B11 #2276, 1:4000), Alexa Fluor secondary antibodies mostly Alexa Fluor Plus 647 (Thermo Fisher Scientific, 1:1000). HCS CellMask Green Stain (Thermo Fisher Scientific, #H32714) for nucleo-cytoplasmic staining was used at 1: 10,000 for 20 min at room temperature.

## Genomic DNA extraction and PCR genotyping

Genomic DNA was extracted either using in-house lysis buffer as described previously (*Bressan et al., 2017*) (bulk populations from 96-well plate or using DNeasy Blood and Tissue Kit (Qiagen, # 69506, for DNA extraction from clonal lines in a 24-well plate). PCR primers flanking the V5 tag were designed online using Primer3Plus to generate 400–600 bp PCR amplicons. PCR genotyping and Sanger sequencing were done as described previously (*Bressan et al., 2017*). DNA samples were analysed using 2.5% agarose gels.

## ChIP-SICAP and RIME

Approximately 40 million mouse NS cells from three T150 flasks (150 cm$^2$) were cultured until 70–80% confluence and then dissociated into single-cells using accutase. To fix DNA-protein and protein-protein interactions, the cell pellet was resuspended in 1.5% methanol-free formaldehyde (Pierce) in 10 mL PBS for 10 min at room temperature. Excess formaldehyde was quenched by adding 125 mM glycine and incubated for 5 min at room temperature. Cells were washed twice with cold PBS and stored at −80°C until further use. ChIP-SICAP experiments were performed as described previously (*Rafiee et al., 2016*). Briefly, chromatin from 40 million formaldehyde-fixed cells was sheared by sonication (Bioruptor Pico, Diagenode) down to 150–500 bp fragments, which were used as input for immunoprecipitation with anti V5 antibody (Abcam, 15828) overnight at 4°C. Antibody was captured with Protein-G beads (LIFE technologies, 10004D), the associated DNA was biotinylated by terminal deoxynucleotidyl transferase (Thermo Fisher, EP0162) in the presence of biotin-11-dCTP (Jena Bioscience, NU-809-BIOX). The antibody was eluted from the beads in 7.5% SDS with 200 mM DTT and the released DNA-protein complexes were caputred by streptavidin magnetic beads (NEB, S1420). After subsequent washes with SDS washing buffer (Tris-CL 1 mM, 1% SDS, 200 mM NaCl, 1 mM EDTA), 20% isopropanol and 40% acetonitrile, the beads were boiled in 0.1% SDS in 50 mM ammonium bicarbonate and 10 mM DTT at 95°C for 20 min. Proteins were digested overnight with trypsin at 37°C and the resulting peptides were purified with the SP3 protocol as described previously (*Hughes et al., 2014*) and analysed using an Orbitrap Fusion LC-MS system.

## crRNA/ssODN design tool

The implementation of our crRNA/ssODN design tool was completed in four stages: extraction of a target genomic sequence from GRCh38.p5 or GRCm38.p4 genome builds, retrieval of crRNA sequences matching the pattern N$_{20}$NGG, scoring and ranking of each crRNA using the 'Rule Set 2' (*Doench et al., 2016*) and 'MIT' scoring models (*Hsu et al., 2013*), and design of each corresponding ssODN sequence.

To accommodate genomic sequence extraction, an SQL database of genomic coordinates was built using the Genomic Features package in R. This SQL database was used to retrieve coding DNA sequence (CDS) ranges upon user query with a desired Ensembl transcript Id. Given a CDS range, a genomic sequence was then extracted from a corresponding GRCh38.p5 or GRCm38.p4 Fasta file.

For each target genomic sequence, crRNAs were extracted limited to the pattern N$_{20}$NGG. crRNAs were then ranked using two scoring models, 'Rule Set 2' for assessing crRNA efficiency, and the MIT scoring system for crRNA specificity (*Hsu et al., 2013*; *Doench et al., 2016*). The former was utilised as a standalone script retrieved from the 'sgRNA Designer' website (https://portals.broadinstitute.org/gpp/public/analysis-tools/sgrna-design) and the latter was implemented as documented on the 'CRISPR Design' website (http://crispr.mit.edu/about). Off targets for each crRNA were found using the short read aligner tool Bowtie, searching up to three mismatches (*Langmead et al., 2009*). Off targets that then match the PAM pattern NAG and NGG were extracted from the Bowtie output. In addition, crRNAs that cut close to the stop codon (8–15 bp in the 3'UTR), and within the UTR region, were prioritised. Given a batch request, the top two ranking crRNAs were selected for output. A maximum distance of 30 bp from the stop codon was chosen as an additional threshold for batch processing.

To implement ssODN design, user-defined tags were inserted immediately 5$^{\text{I}}$ proximal to the stop codon. PAM sequences were changed to minimise potential for Cas9 cleavage of donor sequences. Where the PAM sequence resided in the 3$^{\text{I}}$ UTR, our tool modified the NGG PAM to NGC. Intronic or exonic PAM changes instead aimed to produce silent mutations, and where this was not possible, aimed to reduce alterations in function by minimising differences in hydrophobicity and charge. Final ssODN sequences were limited to 200-mer including the tag sequence. We therefore present the '*Tag-IN*' design tool, a novel crRNA and ssODN design tool aimed at streamlining CRISPR knock-in experimentation (http://tagin.stembio.org).

## Immunoprecipitation-mass spectrometry

The Olig2 protein interactors were identified using the Rapid immunoprecipitation mass spectrometry of endogenous protein (RIME) protocol. The nuclear fraction was resuspended using 1 ml of LB1

(50 mM HEPES KOH pH 7.5, 140 mM NaCl, 1 mM EDTA, 10% glycerol, and 0.5% NP-40) with protease and phosphatase inhibitors. Lysate was cleared by centrifugation at 2000 $g$ for 5 min at 4°C, and the pellet was resuspended with 1 ml of LB2 (10 mM Tris-HCL [pH 8.0], 200 mM NaCl, 1 mM EDTA, and 0.5 mM EGTA), and mixed at 4°C for 5 min. Lysate was cleared by centrifugation and the pellet was resuspended in 0.5 ml of LB3 (10 mM Tris-HCl [pH 8], 100 mM NaCl, 1 mM EDTA, 0.5 mM EGTA, 0.1% Na-deoxycholate, and 0.5% N-lauroylsarcosine). Samples were sonicated in a waterbath sonicator (Diagenode Bioruptor) and cleared by centrifugation. 10 µL V5 trap magnetic beads (MBL) were used per sample. IPs, washes and on-bead digests were performed using a Thermo Kingfisher Duo, all steps are at 5°C unless otherwise stated. Beads were transferred into 500 µL of cleared lysate and incubated for 2 hr with mixing. Beads were then transferred for two washes in RIPA buffer and three washes in non-detergent lysis buffer. On-bead digest was performed by transferring the washed beads into 100 µL 2M urea, 100 mM Tris-HCl pH 7.5, 1 mM DTT containing 0.3 µg trypsin (Promega) per sample, beads were incubated at 27°C for 30 min with mixing to achieve limited proteolysis. The beads were then removed and tryptic digest of the released peptides was allowed to continue for 9 hr at 37°C. Reduced cysteine residues were alkylated by adding iodoacetamide solution to a final concentration of 50 mM and incubated 30 min at room temperature, in the dark. Trypsin activity was inhibited by acidification of samples to a concentration of 1% TFA. Samples were desalted on a C18 Stage tip and eluates were analysed by HPLC coupled to a Q-Exactive mass spectrometer as described previously (*Turriziani et al., 2014*). Peptides and proteins were identified and quantified with the MaxQuant software package (1.5.3.8), and label-free quantification was performed by MaxLFQ (*Cox et al., 2014*). The search included variable modifications for oxidation of methionine, protein N-terminal acetylation, and carbamidomethylation as fixed modification. The FDR, determined by searching a reverse database, was set at 0.01 for both peptides and proteins.

## Acknowledgements

Vectors were provided by Addgene for Cas9 recombinant protein production (Addgene Plasmid 53261). ESCs were provided by Ian Chambers. Vivien for tissue culture, flow-cytometry facility, Bertrand Vernay provided support for microscopy and image analysis. We thank Donal O'Carroll, Ian Chambers, and Abdenour Soufi for helpful comments on the manuscript. The project was also supported by the BBSRC/EPSRC/MRC Synthetic Biological Research Centre (BB/M018040/1), part of the UK Research Councils' investment in 'Synthetic Biology for Growth'. SMP is a Cancer research UK Senior Research Fellow (A17368).

## Additional information

### Competing interests

Ashley M Jacobi, Mark A Behlke: employed by Integrated DNA Technologies (IDT), who sells reagents similar to some described herein. IDT is, however, not a publicly traded company and the authors do not own any shares or equity in IDT. No other authors have any financial interests or relationships with IDT; nor do they own any shares or equity. The other authors declare that no competing interests exist.

### Funding

| Funder | Grant reference number | Author |
|---|---|---|
| Cancer Research UK | A17368 | Pooran Singh Dewari<br>Benjamin Southgate<br>Eoghan O'Duibhir<br>Steven M Pollard |
| Medical Research Council | BB/M018040/1 | Pooran Singh Dewari<br>Steven M Pollard |
| Biotechnology and Biological Sciences Research Council | BB/M018040/1 | Pooran Singh Dewari<br>Steven M Pollard |

| Engineering and Physical Sciences Research Council | BB/M018040/1 | Pooran Singh Dewari Steven M Pollard |
| Brain Tumour Charity | GN-000358 | Pooran Singh Dewari Steven M Pollard |

The funders had no role in study design, data collection and interpretation, or the decision to submit the work for publication.

## Author contributions

Pooran Singh Dewari, Conceptualization, Data curation, Formal analysis, Supervision, Investigation, Methodology, Writing—original draft, Writing—review and editing; Benjamin Southgate, Software, Investigation, Writing—review and editing; Katrina Mccarten, Investigation, Methodology; German Monogarov, Data curation, Formal analysis, Investigation, Methodology; Eoghan O'Duibhir, Niall Quinn, Formal analysis, Investigation, Methodology; Ashley Tyrer, Investigation; Marie-Christin Leitner, Colin Plumb, Maria Kalantzaki, Rebecca Finch, Raul Bardini Bressan, Data curation, Investigation, Methodology; Carla Blin, Resources, Data curation, Methodology; Gillian Morrison, Resources, Funding acquisition, Methodology; Ashley M Jacobi, Resources, Methodology; Mark A Behlke, Conceptualization, Resources, Project administration; Alex von Kriegsheim, Resources, Supervision, Investigation, Methodology, Project administration; Simon Tomlinson, Conceptualization, Software, Supervision, Investigation, Methodology; Jeroen Krijgsveld, Conceptualization, Formal analysis, Supervision, Investigation, Writing—review and editing; Steven M Pollard, Conceptualization, Resources, Supervision, Funding acquisition, Methodology, Writing—original draft, Project administration, Writing—review and editing

## Author ORCIDs

Raul Bardini Bressan (iD) http://orcid.org/0000-0002-5673-9563
Steven M Pollard (iD) http://orcid.org/0000-0001-6428-0492

## Decision letter and Author response

Decision letter https://doi.org/10.7554/eLife.35069.033
Author response https://doi.org/10.7554/eLife.35069.034

# Additional files

## Supplementary files

• Supplementary file 1. List of crRNA targets and ssODN sequences.
DOI: https://doi.org/10.7554/eLife.35069.027

• Supplementary file 2. List of primers for PCR genotyping.
DOI: https://doi.org/10.7554/eLife.35069.028

• Supplementary file 3. Protein interaction partners of Olig2 identified by ChIP-SICAP soluble fraction.
DOI: https://doi.org/10.7554/eLife.35069.029

• Transparent reporting form
DOI: https://doi.org/10.7554/eLife.35069.030

## Data availability

All data generated or analysed during this study are included in the manuscript and supporting files. Newly generated cell lines will be made available on request.

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

## Appendix 1

DOI: https://doi.org/10.7554/eLife.35069.031

# Detailed protocol for epitope knock-in using Cas9 RNP complex

This protocol describes the step-by-step procedure for epitope tagging in neural stem (NS) cells. The same protocol can be used for other cell lines including ES cells. We used an Amaxa 4D Nucleofection system for the delivery of Cas9 RNP. The typical nucleofection reaction volume is 20 µL (Lonza transfection buffer) plus 5 µL (CRISPR/Cas9 reagents) in a 16-well microcuvette. Calculations described below are for one nucleofection reaction and can be easily scaled up depending on the number of transfections.

## 40–45 min before transfection: csRNP preparation

(1) In a sterile 0.2 mL PCR tube, mix 1.1 µL of crRNA and 1.1 µL of tracrRNA (each stock solution 100 µM). Vortex and spin down.

(2) Anneal cr:tracrRNAs on PCR block using the following program:
- 95°C for 5 min
- 95°C – 25°C: step cool-down at ramp-rate 0.1°C/sec
- 4°C (store) at ramp rate 0.5°C/sec

Alternatively, heat the cr:tracrRNA mixture at 95°C for five minutes and let it cool at room temperature for 30 min and plunge into ice.

(3) To the annealed cr:tracrRNAs, add 10 µg of recombinant Cas9 protein (with NLS and at 5–10 µg/µL stock concentration). Ideally, the volume of these two components should be no more than 5–7 µL. Incubate at room temperature for 10 min to allow Cas9 RNP formation and then plunge into ice until transfection.

## Just before transfection: Prepare cells for transfection

Pre-warm 2 mL complete media supplemented with EGF/FGF/Laminin in a 6- or 12-well plate. Cells will be transferred into this immediately after transfection.

(4) Harvest cells using accutase dissociation. Count cells using trypan blue dye to ignore dead cells. Transfer $1.5 \times 10^5$ NS cells per transfection in a 15 mL tube and centrifuge cells at 1400 rpm for 3 min.

(5) Discard the supernatant and re-suspend the cells in 20 µL of supplemented SG cell line buffer (SG cell line buffer should be brought to room temperature before use). Mix gently with pipette, avoid bubbles. Transfer to the 0.2 mL PCR tube that contains the Cas9 RNP.

Note: Other cell types may require optimization of Lonza nucleofection buffer for high viability and transfection efficiency.

(6) Add 1 µL of ssODN (30 µM stock in IDT Duplex Buffer) to the cell suspension and Cas9 RNP mixture. Mix gently by pipetting, avoid bubbles.

## Electroporation

(7) Set up the nucleofection program on 4D Amaxa machine to DN-100 for mouse NS cells.

(8) Carefully transfer cell suspension/Cas9 RNP mixture at the centre of a microcuvette well, avoid bubbles. We use two consecutive pulses of DN-100 program for mouse NS cells.

(9) Immediately after the nucleofection pulse(s), remove the microcuvette strip and add 150 µL of pre-warmed supplemented media onto each microcuvette well. Transfer the cells into a pre-warmed 6 or 12-well plate containing 2 mL of supplemented media.

(10) Let the cells grow for 3–5 days, assay tag knock-in using ICC or PCR.

## Downstream assays

### ICC

Seed $1 \times 10^4$ cells in a 96-well plate or $2 \times 10^4$ cells in a 48-well plate for immunostaining. For the V5 tag detection, we use eBioscience anti-V5 tag antibody (1:1000) overnight at 4℃. We prefer the far-red channel for secondary antibody (Alexa 647), as it gives less background and is brighter as compared to the green channel.

anti-V5 tag antibody: https://www.thermofisher.com/antibody/product/V5-Tag-Antibody-clone-TCM5-Monoclonal/14-6796-80

### PCR

We use a nearly confluent 96- or 48-well plate for crude DNA extraction. Wash cells and add 50 µL of lysis buffer (0.45% NP40, 0.45% Tween 20, 200 µg/mL Proteinase K, 0.5x NEB LongAmp buffer) directly into well; no accutase required. Incubate plate at 37℃ for 1 hr, transfer the DNA extract into 0.2 ml PCR tubes and incubate at 50℃ for 2 hr. Deactivate PK by heating at 95℃ for 10 min. Use 2 µL of the final DNA extract as a template for 20 µL LongAmp PCR reaction. Store DNA at −20℃. To assess tag knock-in in the bulk populations, we use an exon-specific forward primer (400–500 bp upstream of the Stop codon) and a V5 tag-specific reverse primer. For the clonal lines, we use primers flanking the V5 tag such that the product size is ~400 bp in the parental cells and ~450 bp after the V5 knock-in.

LongAmp taq DNA polymerase: https://www.neb.com/products/m0323-longamp-taq-dna-polymerase

## PCR set up for 20 µL reaction volume

- Sterile water - 10.8 µL
- NEB 5x LongAmp buffer - 4.0 µL
- 10 mM dNTPs- 0.8 µL
- 10 µM For primer- 0.8 µL
- 10 µM Rev primer- 0.8 µL
- Taq polymerse- 0.8 µL
- Template- 2.0 µL (100 ng)

## PCR conditions

- Step1: Initial Denaturation 95℃ - 1 min
- Step2: 38 cycles
- 94℃ for 30 s
- 50-60℃ - 30 s
- 65℃ - 45 s (for 600 bp PCR product)
- Step 3: Final extension
- 65℃ - 10 min
- Hold at 4℃.

Analyze PCR products on a 2.5% agarose gel, use 1 kb$^+$ gene ruler DNA ladder for size estimation.

