## [Decision Letter]

Thank you for submitting your article "An efficient and scalable pipeline for epitope tagging in mammalian stem cells using Cas9 ribonucleoprotein" for consideration by *eLife*. Your article has been reviewed by two peer reviewers, and the evaluation has been overseen by a Reviewing Editor and Fiona Watt as the Senior Editor. The following individual involved in review of your submission has agreed to reveal his identity: Hein te Riele (Reviewer #1).

The reviewers have discussed the reviews with one another and the Reviewing Editor has drafted this decision to help you prepare a revised submission.

Summary:

Dewari et al., present a pipeline for high content epitope tagging of proteins in mammalian cells. Salient features of their approach are: (i) the use of purified CRISPR/Cas9 protein loaded with a 67-mer tracrRNA and a gene-specific 36-mer crRNA to introduce a double-strand break in the 3' UTR of the target gene in close vicinity of the stop codon. The RNP is introduced into cells by electroporation. In Figure 1 they show that the synthetic cr:tracr dual RNA system outperforms in vitro transcribed single guide RNAs. (ii) for homology-directed break repair, they use {plus minus}200 nucleotide single-stranded oligo deoxyribonucleotide templates that instruct the integration of the epitope tag. To prevent recutting, they simultaneously introduce a PAM-disrupting mutation. The frequency of successful tagging is around 15-20%, occasionally even 30%.

Overall, this report is very well written. The figures support the conclusions of the paper and present a convenient guideline for tagging of proteins with small epitopes. As such, the manuscript may be of interest for a broad audience. However, some points need further clarifications.

Essential revisions:

1) The frequency of successful epitope tagging was calculated based on the percentage of cells expressing the epitope (Figure 1). Subcloning and subsequent PCR analysis of V5-positive clones revealed many clones with bi-allelic tagging (Figure 2B, Table 1). However, this analysis also showed not all clones were correctly targeted. E.g., lane 1 of the *Sox2*-V5 clones and lanes 3 and 6 of the Sox3-V5 clones (Figure 2B). It would be helpful if the authors would add to Table 1 the number of non-correctly targeted clones in order to more precisely determine the targeting efficiency.

2) In subsection “The high frequency of knock-in using csRNP facilitates simple recovery of biallelictagged clonal lines”, reference is made to Table 2. This apparently should be Table 1. Or Supplementary table 2, which is the same as Table 1?

3) Figure 3C shows that epitope tagging can also be achieved at genes that are not expressed in ES cells. Only after differentiation into neural cells, the V5 tag became visible. However, it seems as if in the differentiated ES cells all neuronal cells expressed V5. How is this possible? Were neural rosettes obtained from single ES cells? Were also V5-negative neuronal rosettes obtained. If so, could an estimation be given on the frequency of epitope tagging at non-expressed genes versus expressed genes in ES cells by comparing the number of V8-negative and -positive rosettes?

4) Figure 4D shows the fusion between two target genes and the V5 tag. However, it is unclear why this information is useful as the fusion is present in the ssODN template. It would be more interesting to see how the sequence around the DSB was repaired.

Similar data is shown in Figure 1—figure supplement 1, Figure 2—figure supplement 1, Figure 3—figure supplement 1 and Figure 4—figure supplement 3. From a skeptical point of view, these sequences only show that the ssODNs were properly synthesized. To confirm "in-frame and error-free insertion of V5 tag sequence at the C-terminus", mRNA sequencing appears more appropriate to show the tag is really present at the destined site. Reassuringly, Figure 1—figure supplement 2BFigure just shows the modified *Sox2* PAM site. Anyway, it is important to indicate that false positives were excluded by properly choosing the position of PCR primers.

5) The higher performance of gRNAs cutting proximal to the insertion site of the epitope tag compared to distal cutting sites shown in Figure 4 is very interesting and may relate to suggested mechanisms of ssODN-templated DSB repair. It would be interesting if the authors could share their opinion on the reasons for the inferior performance of distal gRNAs.

6) The higher performance of tag-proximal gRNAs was only seen by staining for the V5 epitope (Figure 4), not by PCR (Figure 4—figure supplement 1, Figure 4—figure supplement 2), according to the authors because the latter is not quantitative. It is not clear why in Figure 4—figure supplement 1, Figure 4—figure supplement 2 some lanes are indicated +, others not. Was this only based on the expected size of the PCR product? Please indicate this in the legend. How discordant were the PCR and ICC results?

7) Related to the above: In subsection “Design of guide RNAs and ssODN repair templates” it is mentioned that "the 200-mer ssODN ultramer was chosen to be the same strand as the guide RNA". Was there a particular reason to choose this strand and not the opposite strand? However, "for some of the ultramers the PAM-strand synthesis had failed and therefore, the complementary strand (non-PAM strand) was synthesized as a donor DNA." It would be very interesting to know how these non-PAM ssODNs performed. Do the authors consider PAM and non-PAM ssODNs equally effective? Has a comparison been made between the two?

8) The above is important with respect to the Tag-IN tool the authors have created. Does the tool always propose a PAM ssODN or can also the reversed strand be chosen? If so, it should be made clear form comparative experiments that the two strands perform equally well.

9) In subsection “A scalable pipeline for high-throughput knock-in of epitope tags using 96-well Microplates”, the results of simultaneous tagging of 185 genes is presented. Plate 1 (Figure 6A) and 2 (Figure 6C) show 30/90 and 31/95 TF2 positive for V5. What was the cutoff here? According to the scale bar, more positive wells are visible with low ({plus minus}2-5%) but detectable V5 signal. On the other hand, what happened to the negative wells ({plus minus}77 in total)? Are these all non-expressed genes? This should be made clear in order to estimate how effective the Tag-IN tool works.

10) What would the authors recommend for tagging proteins with larger tags, such as GFP or luciferase? Would RNPs work similar well with larger double- or single-stranded templates?

11) A large number of stem and progenitor lines have been generated that would be of great value as a resource for the scientific community. Would it be possible to make them available over a cell repository?

12) The presented Tag-IN website was not available online. Thus, the value of it can't be judged.

Although likely only a small glitch, there should be measures to ensure that the online tool will be active for years to come after publication.

13) With regard to the tools and resource section selection, addition of a detailed supplementary protocol would be helpful

---

## [Author Response]

Summary:Dewari et al., present a pipeline for high content epitope tagging of proteins in mammalian cells. Salient features of their approach are: (i) the use of purified CRISPR/Cas9 protein loaded with a 67-mer tracrRNA and a gene-specific 36-mer crRNA to introduce a double-strand break in the 3' UTR of the target gene in close vicinity of the stop codon. The RNP is introduced into cells by electroporation. In Figure 1 they show that the synthetic cr:tracr dual RNA system outperformsin vitro transcribed single guide RNAs. (ii) for homology-directed break repair, they use {plus minus}200 nucleotide single-stranded oligo deoxyribonucleotide templates that instruct the integration of the epitope tag. To prevent recutting, they simultaneously introduce a PAM-disrupting mutation. The frequency of successful tagging is around 15-20%, occasionally even 30%.Overall, this report is very well written. The figures support the conclusions of the paper and present a convenient guideline for tagging of proteins with small epitopes. As such, the manuscript may be of interest for a broad audience. However, some points need further clarifications.Essential revisions:1) The frequency of successful epitope tagging was calculated based on the percentage of cells expressing the epitope (Figure 1). Subcloning and subsequent PCR analysis of V5-positive clones revealed many clones with bi-allelic tagging (Figure 2B, Table 1). However, this analysis also showed not all clones were correctly targeted. E.g., lane 1 of the Sox2-V5 clones and lanes 3 and 6 of the Sox3-V5 clones (Figure 2B). It would be helpful if the authors would add to Table 1 the number of non-correctly targeted clones in order to more precisely determine the targeting efficiency.

Table 1 has been now updated to display the number and percentage of correctly targeted clones as determined by PCR genotyping.

2) In subsection “The high frequency of knock-in using csRNP facilitates simple recovery of biallelictagged clonal lines”, reference is made to Table 2. This apparently should be Table 1. Or Supplementary table 2, which is the same as Table 1?

We have fixed this. Reference is made to the correct Table 1 and Supplementary table 2 has been removed.

3) Figure 3C shows that epitope tagging can also be achieved at genes that are not expressed in ES cells. Only after differentiation into neural cells, the V5 tag became visible. However, it seems as if in the differentiated ES cells all neuronal cells expressed V5. How is this possible? Were neural rosettes obtained from single ES cells? Were also V5-negative neuronal rosettes obtained. If so, could an estimation be given on the frequency of epitope tagging at non-expressed genes versus expressed genes in ES cells by comparing the number of V8-negative and -positive rosettes?

This is a good suggestion. We realise this was not clearly explained in the results and has now been updated. Rosettes were obtained from the bulk populations and not the clonal ES cell lines. Both V5-positive and V5-negative rosettes were obtained. In the uniformly tagged rosettes these were likely generated from the expansion of single tagged ES cells during the mixed differentiation.

We have now calculated the approximate knock-in efficiency by counting V5-positive colonies out of the total Nestin-expressing colonies. The knock-in efficiency for both expressed and non-expressed genes in ES cells have been added in the Results section; these were found to be at a similar level to NS cells.

4) Figure 4D shows the fusion between two target genes and the V5 tag. However, it is unclear why this information is useful as the fusion is present in the ssODN template. It would be more interesting to see how the sequence around the DSB was repaired.Similar data is shown in Figure 1—figure supplement 1, Figure 2—figure supplement 1, Figure 3—figure supplement 1 and Figure 4—figure supplement 3. From a skeptical point of view, these sequences only show that the ssODNs were properly synthesized. To confirm "in-frame and error-free insertion of V5 tag sequence at the C-terminus", mRNA sequencing appears more appropriate to show the tag is really present at the destined site. Reassuringly, Figure 2—figure supplement 1B just shows the modified Sox2 PAM site. Anyway, it is important to indicate that false positives were excluded by properly choosing the position of PCR primers.

The majority of the gRNAs selected cut in the 3’ UTR. For PCR genotyping, of bulk populations, we used a forward primer designed flanking the left homology arm (i.e. it was not present within the ssODN, but the upstream exon sequence) (Figure 1—figure supplement 1, Figure 3—figure supplement 1 and Figure 4—figure supplement 3). This product therefore confirms correct in-frame insertion. We have updated the Figure legend to make this clear. Indels in the 3’UTR in the bulk population could not be assessed by Sanger sequencing using this PCR product; however, for the clonal lines we used PCR primers flanking the V5 tag (Figure 2—figure supplement 1).

The figure has now been updated to clearly show the sequences around the guide RNA cut-site and confirms the error-free knock-in of V5 tag sequence.

5) The higher performance of gRNAs cutting proximal to the insertion site of the epitope tag compared to distal cutting sites shown in Figure 4 is very interesting and may relate to suggested mechanisms of ssODN-templated DSB repair. It would be interesting if the authors could share their opinion on the reasons for the inferior performance of distal gRNAs.

We have now added another reference (Guo et al., 2018) relevant to this observation. In the Discussion section we now speculate that the “partial HDR” could be the probable reason for the low knock-in efficiency achieved with the distal-cutting set of gRNAs.

6) The higher performance of tag-proximal gRNAs was only seen by staining for the V5 epitope (Figure 4), not by PCR (Figure 4—figure supplement 1, Figure 4—figure supplement 2), according to the authors because the latter is not quantitative. It is not clear why in Figure 4—figure supplement 1, Figure 4—figure supplement 2 some lanes are indicated +, others not. Was this only based on the expected size of the PCR product? Please indicate this in the legend. How discordant were the PCR and ICC results?

Lanes marked with ‘+’ denote successful knock-in of the V5-encoding sequence at the target gene. The legends for both the Figure 4—figure supplement 1 and Figure 4—figure supplement 2 have been updated to make this clear.

Genes positive for V5 knock-in as assessed by ICC were positive by PCR genotyping. However, many expressed-genes (e.g. *Sox5, Foxj3, Foxk1*) were positive by PCR for the second set of gRNAs (distal-cutting), but we only detected a few V5-positive cells in the entire microscopic field. The increased sensitivity of PCR therefore likely explains this difference. V5 immunocytochemistry remains in our view the simplest way to confirm successful knock-in.

7) Related to the above: In subsection “Design of guide RNAs and ssODN repair templates” it is mentioned that "the 200-mer ssODN ultramer was chosen to be the same strand as the guide RNA". Was there a particular reason to choose this strand and not the opposite strand? However, "for some of the ultramers the PAM-strand synthesis had failed and therefore, the complementary strand (non-PAM strand) was synthesized as a donor DNA." It would be very interesting to know how these non-PAM ssODNs performed. Do the authors consider PAM and non-PAM ssODNs equally effective? Has a comparison been made between the two?

Gasiunas et al., 2012 (PNAS) had suggested that the Cas9 complex can cut the complementary ssDNA (non-PAM strand) regardless of PAM presence. We initially had chosen the non-complementary strand of the DNA (PAM ssODN) for donor DNA synthesis as a precautionary step to avoid risks of ssODN being cut during in vitro assembly of CRISPR ingredients. However, a recent report suggests that for symmetrical PAM ssODNs there is no strand bias in terms of HDR knock-in efficiency (Liang et al., 2017). We have updated the Discussion section to make clear that our study focussed on the PAM ssODN, but in light of these recent publications the alternative strand will likely also be effective.

8) The above is important with respect to the Tag-IN tool the authors have created. Does the tool always propose a PAM ssODN or can also the reversed strand be chosen? If so, it should be made clear form comparative experiments that the two strands perform equally well.

Our tool designs only the PAM ssODNs as a donor DNA. The user can easily order the reverse complement sequence if they wish to test both.

9). In subsection “A scalable pipeline for high-throughput knock-in of epitope tags using 96-well Microplates”, the results of simultaneous tagging of 185 genes is presented. Plate 1 (Figure 6A) and 2 (Figure 6C) show 30/90 and 31/95 TF2 positive for V5. What was the cutoff here? According to the scale bar, more positive wells are visible with low ({plus minus}2-5%) but detectable V5 signal. On the other hand, what happened to the negative wells ({plus minus}77 in total)? Are these all non-expressed genes? This should be made clear in order to estimate how effective the Tag-IN tool works.

For the V5 tagging of 185 genes in the 96-well format, we used high-content imaging with Operetta to calculate the knock-in efficiency. The success of V5 tag insertion was assessed by manually inspecting images from each of the 96 wells and those with clear nuclear staining were scored as positive (30/90 and 31/95 both plates). The gRNAs for the 185 genes that were designed using Tag-IN tool were also verified manually to confirm that they were cutting near the insertion site (proximal-cutting) and had high score.

RNA-seq data we have recently obtained from other NS cell lines suggests the majority of the 185 genes are expressed – but we found that 10-20% are either not detectable or expressed at low levels. So, our ~30% of tagging success is likely a significant underestimate. For some, however, the failure is likely due to either a poor gRNA, poor repair ssODN or both. We now make this clear in the Discussion section.

10) What would the authors recommend for tagging proteins with larger tags, such as GFP or luciferase? Would RNPs work similar well with larger double- or single-stranded templates?

This is a good suggestion and of great interest to our lab. Over the recent months we have attempted to deploy the RNPs with synthetic dsDNA blocks encoding fluorescent proteins. We are happy to find that this works very well for Olig2, *Sox2* and Foxg1. We see 1% efficiency (as anticipated, less than V5 tags; but still sufficient to recover cell lines by FACs). We include these data as a new supplementary Figure 7, which shows the knock-in of mCherry reporter at the C-terminus of mouse *Sox2*, Olig2, and Foxg1 loci alongside some testing of different sized arms (we find that ~500bp is optimal for efficiency and costs of synthesis).

11) A large number of stem and progenitor lines have been generated that would be of great value as a resource for the scientific community. Would it be possible to make them available over a cell repository?

Yes. We have already distributed many lines. We have put a note in the methods that all lines and tagged derivatives are available on request.

12) The presented Tag-IN website was not available online. Thus, the value of it can't be judged.Although likely only a small glitch, there should be measures to ensure that the online tool will be active for years to come after publication.

The website is now available using the following login: http://tagin.stembio.org

We have now made this tool live to the public. The tool will be supported/hosted at the MRC Centre for Regenerative Medicine, Edinburgh. There is funding in place from Cancer Research UK to support this for at least the next six years (it will be linked to a new ‘Glioma cellular genetics resource’).

13) With regard to the tools and resource section selection, addition of a detailed supplementary protocol would be helpful.

We agree this would be useful. We have now included a more step-by-step protocol for users (Supplementary Protocol).